



# RADAR-Derived Precipitation Climatology for Wind Turbine Blade Leading Edge Erosion

**Frederick Letson[1] (ORCID: 0000-0001-9275-0359), Rebecca J. Barthelmie[2] (ORCID: 0000-0003-0403-6046), Sara C. Pryor[1] (ORCID: 0000-0003-4847-3440)**

[1]Department of Earth and Atmospheric Sciences, Cornell University, Ithaca, New York

[2]Sibley School of Mechanical and Aerospace Engineering, Cornell University, Ithaca, New York

*Correspondence to*: F. Letson (fl368@cornell.edu) and S.C. Pryor (sp2279@cornell.edu)

**Abstract:** Wind turbine blade leading edge erosion (LEE) is a potentially significant source of revenue loss for windfarm
operators. Thus, it is important to advance understanding of the underlying causes, to generate geospatial estimates of erosion
potential to provide guidance in pre-deployment planning and ultimately to advance methods to mitigate this effect and extend
blade lifetimes. This study focusses on the second issue and presents a novel approach to characterizing the erosion potential
across the contiguous USA based solely on publicly available data products from the National Weather Service dual-polarization
RADAR. The approach is described in detail and illustrated using six locations distributed across parts of the USA that have
substantial wind turbine deployments. Results from these locations demonstrate the high spatial variability in precipitation-
induced erosion potential, illustrate the importance of low probability high impact events to cumulative annual total kinetic
energy transfer and emphasize the importance of hail as a damage vector.

## 1    Introduction and objectives

In 2017 wind turbines (WT) provided 6% of total electricity generation in the United States of America (USA) (U.S.
Energy Information Administration, 2018) and there are over 50,000 WT operating in the USA today (Pryor et al., 2019). WT
are subject to harsh operating conditions during their 20-25 year lifetimes including; extreme winds, impacts from heavy rain,
hailstones and snow, and intense ultraviolet light exposure that can lead to materials damage (Keegan et al., 2013). Accordingly,
operation and maintenance (O&M) costs comprise 20-25% of the total levelized cost per kWh of electricity produced over the
WT lifetime (Mishnaevsky Jr, 2019;Moné et al., 2017). WT blades exhibit the highest failure rate (FR ~ 0.2) of any WT
component (Zhu and Li, 2018). The most expensive repair and longest repair times are associated with blades (Shohag et al.,
2017). Estimates suggest average cost of blade repair is approx. $30,000, with replacement costs of ~ $200,000 (Mishnaevsky Jr,
2019).

A key cause of the need for blade repairs is excess damage (i.e. material loss) on the leading edge (leading edge erosion,
LEE). LEE roughens WT blades, reducing lift and electrical power production (Sareen et al., 2014;Gaudern, 2014). LEE causes
an average of 1-5% reduction in annual energy production (AEP) (Froese, 2018) and up to a 9% reduction when delamination
occurs (Schramm et al., 2017). Thus, excess LEE may be costing the industry tens of millions of dollars per year via lost revenue
and/or increased maintenance costs, and poses a threat to achieving continuing wind energy cost reductions (Sareen et al., 2014).
In response to this issue a major industrial research consortium from Europe (including DNV GL, Vestas and Siemens Gamesa
Renewable Energy) has recently (Nov 2018) announced a new partnership (COBRA) focused on analysis of mitigation measures
for LEE including development of next-generation leading-edge protection systems (Durakovic, 2019).





WT blades use composites (e.g. epoxy or polyester, with reinforcing glass or carbon fibers) (Mishnaevsky et al., 2017) coated to protect the blade structure by distributing and absorbing the energy from impacts (Brøndsted et al., 2005). Thus, the leading edge actually comprises several layers of the main structural composite material (and thickening materials) plus coatings (Mishnaevsky et al., 2017). Impact fatigue caused by collision with rain droplets and hail stones is a primary cause of WT blade

LEE (Bech et al., 2018;Bartolomé and Teuwen, 2019;Zhang et al., 2015). Although rain droplets fall at only modest velocities (typically $\leq 10$ ms$^{-1}$, see details below), the tip of WT blades rotate quickly (50-110 ms$^{-1}$), thus the net closing velocity and kinetic energy transfer are large. Each precipitation impact on the blade leading edge results in transient stresses that are proportional to impact velocity (Preece, 1979;Slot et al., 2015). The stress induced by individual high net collision impacts with hydrometeors may, in principle, exceed the strength of the material. Estimates of the failure energy threshold of a composite

structure vary widely (e.g. values of $72 - 140$ J are given in (Appleby-Thomas et al., 2011)) and may exceed 300 J for leading-edge thicknesses and hailstone diameters > 20 mm (Kim and Kedward, 2000). However, conceptually the erosion of homogeneous materials is most frequently considered using a three stage model. Initially there is an incubation period during which impacts occur but no visible damage is observed although microstructural changes in the materials generate nucleation sites for material removal which commences when a threshold is reached (i.e. when some level of accumulated impacts is

reached). Once the time to damage has been exceeded additional damage occurs as stress waves propagate from the impact sites into the composite and cause existing pits and cracks to grow and there is a steady increase of material loss occurs with each additional impact (Cortés et al., 2017;Eisenberg et al., 2018;Traphan et al., 2018). The number of impacts required to reach the threshold for surface fatigue failure is a function of the droplet diameter and phase, the closing velocity, the strength of the material and the pressure of the impact. Hence, the materials response to hail (solid hydrometeors) may differ from that to

collisions with liquid (rain) droplets. For example, the maximum von-Mises stress created in the WT blade leading edge from a 10 mm diameter hailstone greatly exceed that from a rain droplet of equivalent size and closing velocity due to differences in mass and hardness (Keegan et al., 2013).

WT LEE is a developing area of research and uncertainty remains regarding frequency and severity of the issue. Rates of LEE appear to be highly spatially variable due to variations in WT operating conditions and the precipitation climate. Industrial

experience has demonstrated exposure to particularly harsh operating conditions can erode coatings causing partial delamination after as little as 2-3 years (Rempel, 2012;Keegan et al., 2013). Elastomeric coatings can be applied for additional erosion resistance (Dalili et al., 2009;Valaker et al., 2015). However, the life of such coatings cannot be predicted accurately (and is a function of UV exposure, (Shokrieh and Bayat, 2007)), they have a negative impact on blade aerodynamics (Giguère and Selig, 1999) and their cost-effectiveness is uncertain (Dashtkar et al., 2019).

The total installed capacity (IC), rated capacity (and physical dimensions) of wind turbines (WT) being installed exhibited marked growth in the USA over the last 20 years (Wiser and Bolinger, 2018;Wiser et al., 2016). Average WT blade length increased from < 4 m in 1985 to 32 m in 2005 and now exceeds 55 m (Wiser and Bolinger, 2018). Since the tip speed increases with blade length, this tendency towards taller WT with longer blades exacerbates LEE potential. Based on previous research the *a priori* expectation of this research is that excess LEE is most likely on WT deployed in environments with high rain intensities

and hail frequencies such are experienced in the Central Plains (Fig. 1). LEE is likely to present a growing issue within the US wind industry as more and larger wind turbines with higher tip-speed ratios are deployed (Amirzadeh et al., 2017b). The current average age of WT in the US is 9 years (AWEA, 2019) and LEE will be of greater concern as a larger number of WT move out of the typical 1 to 5 year warranty period (Bolinger and Wiser, 2012;Brown, 2010).

Addressing the challenges posed by blade LEE and developing mitigation options requires multi-scale and multi-disciplinary

research. Given the importance of precipitation phase, size and intensity during WT operation to the potential for blade LEE here





we focus on developing a consistent and generalizable framework that can be applied to derive estimate of erosion-relevant atmospheric properties. We present an objective, spatially consistent, robust and repeatable framework that can be applied across the continental USA and crucially uses only non-commercial (i.e. publicly available) data. The specific objectives of the research reported herein are:

1) To develop the workflow necessary to develop a proto-type RADAR-based erosion atlas.

2) To provide a first estimate of the spatial variability of erosion potential across CONUS in regions where wind turbines are currently deployed (see Fig. 1).

3) To conduct an initial uncertainty propagation exercise to illustrate how uncertainties in the input data propagate through the analysis workflow to influence erosion potential estimates.

4) To describe the degree to which blade LEE is episodic and therefore amendable to the mitigation strategy proposed earlier in research from Denmark of WT curtailment during 'highly erosive' periods (Bech et al., 2018).

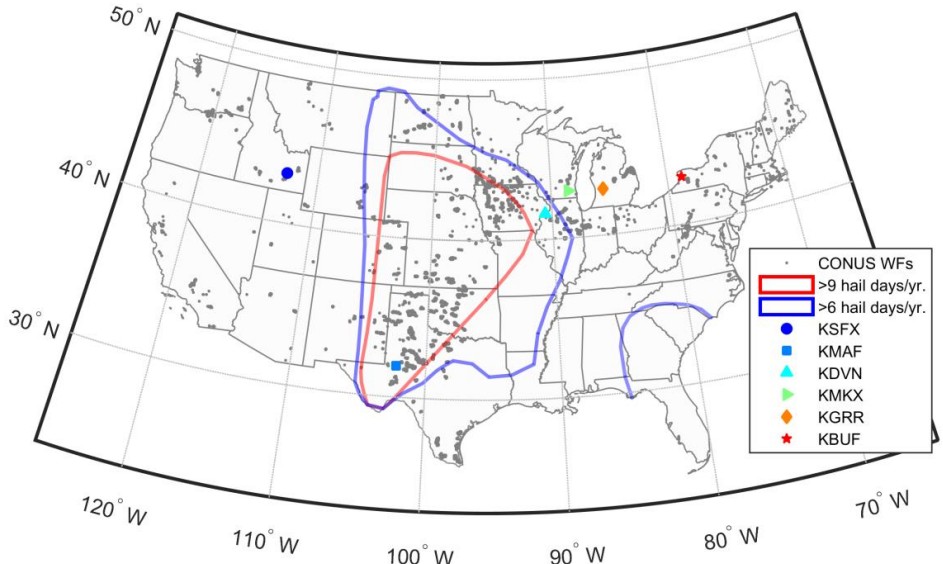

**Figure 1 Locations of wind turbines as deployed at the end of 2017 according the UGSGS database (available from; https://eerscmap.usgs.gov/uswtdb/) (grey dots), the NWS RADAR stations from which data are presented (see details in Table 1), and**

**areas of frequent hail occurrence. Areas with more than nine hail days per year are outlined by the red contour, and those with more than six are outlined by the blue contours (Cintineo et al., 2012).**



**Table 1: The station code and locations of the six NWS dual polarization RADARs from which data are presented (listed from west to east).**

| Station code | Latitude (N) | Longitude (E) | State |
|---|---|---|---|
| KSFX | 43.106 | -112.686 | ID |
| KMAF | 31.943 | -102.189 | TX |
| KDVN | 41.612 | -90.581 | IL |
| KMKX | 42.968 | -88.551 | WI |
| KGRR | 42.894 | -85.545 | MI |
| KBUF | 42.949 | -78.737 | NY |

## 2    Data and Methods

A first estimate of precipitation-derived erosion potential at sites across the USA as developed in the current work is based on a characterization of the kinetic energy exchange from rain and hail impacts on the blade leading edge. The procedure used in making these estimates is divided into two steps: Calculation of meteorological parameters (wind speed, rain and hail) at a nominal wind farm located within the observation areas of six RADARs and then calculation of blade impact frequencies and
energy transfer based on those meteorological parameters.

The research reported herein leverages resources generated from the upgraded National Weather Service (NWS) network of WSR-88D RADAR to dual polarization (completed in 2013, (Seo et al., 2015;Crum et al., 1998)) along with the NOAA Weather and Climate Toolkit (WCT) (see details of the data products and data volumes provided in Appendix A). These data represent a unique opportunity to characterize precipitation properties such as hail that are very challenging to detect and to accurately
characterize using in situ methods or human observers (see discussion in (Allen and Tippett, 2015) and details of RADAR operation (Kumjian, 2018)). NWS RADAR operate at elevation angles between 0.5° and 19.5° and an azimuthal resolution of 1°. Doppler and dual-polarization data are publicly available at a resolution of 0.25 km up to a range of 300 km from each RADAR site (NOAA, 1991;Istok et al., 2009) (see description of the data provision in (Kelleher et al., 2007) and an example of the NWS products given in Fig. 2). The temporal resolution of the data is typically ~ 5 minutes, but varies slightly with scanning mode: 1)
Clear Air Mode uses longer, 10-minute scans to collect sufficient return data during times of no precipitation when signal return strength is relatively low. 2) Precipitation mode is used when there is any precipitation detected in the scan area and uses a 6-minute scan cycle. 3) Storm mode is used when severe or rapidly-evolving storms are present, and uses a 5-minute sampling interval, made possible by reducing the number of elevation angles used (NOAA, 2016a). Storm detection and tracking using RADAR is a complex and evolving science but in brief the NWS system uses an automated function which employs reflectivity
from the current scan and storm cell location and vertically integrated liquid water (VIL) from the previous scan (Johnson et al., 1998).

To illustrate the proposed analysis framework we use data from six NWS dual-polarization Doppler RADAR stations (see Fig. 1 and Table 1) collected over the period 2014-2018. These locations were chosen to represent gradients in hail probability and precipitation amount in regions with relatively high wind turbine installed densities (Fig. 1). We employ the framework in
order to generate erosion climates for six nominal wind farms operating in the scanned volume of the RADARs and located 35-75 km from the RADAR locations. The following RADAR data products are used (see also Appendix A):



- Hourly precipitation rate (N1P) is the precipitation rate in each RADAR cell in mmhr$^{-1}$ as estimated from reflectivity.

- Hybrid Hydrometer Classification (HHC): Based on reflectivity, temperature, and dual polarization variables, HHC is an estimate of the most likely targets within the RADAR volume (NOAA, 2016b;Chandrasekar et al., 2013). The hydrometeor types encoded in the NWS data product are; dry snow, wet snow, crystals, big drop, rain (light and moderate), heavy rain, graupel, and rain with hail.

- Hail reports (NHI): Maximum hail size (an estimate of the 75th percentile hail stone diameter ($D_{75}$)) and probability of hail are used to identify the occurrence and severity of hail events (see discussion in (Witt et al., 1998)).

- Composite Reflectivity (NCR): Maximum reflectivity at any elevation angle measured in each RADAR cell. This is used here to characterize the spatial extent of hail events (i.e. reflectivity > 50 dBZ (Witt et al., 1998)).

- Radial wind speeds from the 0.5° elevation angle as computed from the Doppler shift (N0V) (Alpert and Kumar, 2007).

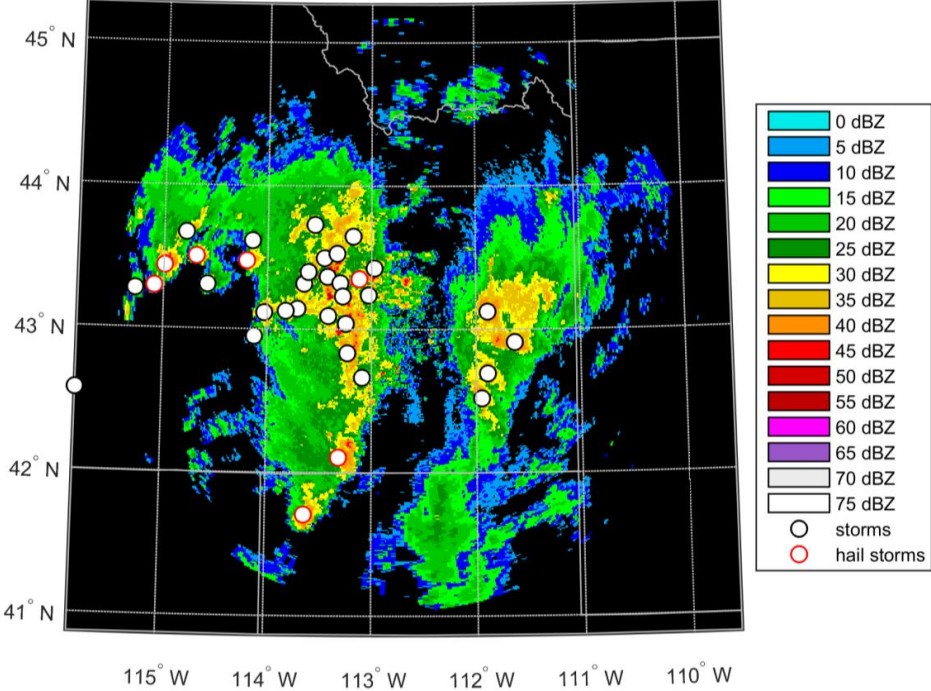

**Figure 2 – Example of a single five minute period of RADAR data from KSFX (August 8, 2013, 22:37 UTC). The colors show the composite reflectivity (i.e. the maximum reflectivity from any of the elevation angles sampled by the NWS RADAR) in dBZ. The circles represent storm cells that are identified and tracked by the NWS detection algorithm; black circles are storms without hail, and red circles are those with hail.**

Wind speeds, hydrometeor type and precipitation intensity for each nominal wind farm located within each RADAR scanned area in each five minute period (Fig. 3) are derived as follows:

Precipitation intensity is characterized by rainfall rate ($RR$) in mmhr$^{-1}$, which is derived using RADAR Z-R relationships (Wilson and Brandes, 1979) and reported in the parameter precipitation rates (N1P) in all RADAR cells. A spatial mean of all N1P values in all RADAR cells within 5 km of the nominal wind farm centroid is used here. This rainfall rate is also



used to derive the raindrop spectrum using the Marshall-Palmer distribution (Marshall and Palmer, 1948). In it the number of droplets above radius, $R$, per cubic meter of air ($N$, m$^{-3}$) is given by;

$$N = \frac{N_0}{\Lambda} e^{-\Lambda R} \qquad (1)$$

Where $\Lambda = 8200(RR)^{-0.21}$ (m$^{-1}$), $RR$ is the rainfall rate in mmhr$^{-1}$, and $N_0 = 1.6 \times 10^7 m^{-4}$. See an example rain droplet size distribution, expressed as $dN/dR$, for a $RR$ of 25 mmhr$^{-1}$ in Fig. 3.

Hail occurrence is characterized by a number of NWS RADAR-derived parameters, most of which are contained in the hail reports (NHI). Hail size and probability of occurrence are conservatively estimated here, by taking the largest values reported for any storm cell within 5 km of the nominal wind farm centroid. The spatial coverage of hail within that 5 km radius is determined by calculating the fraction of RADAR cells in that area which have composite reflectivity in excess of 50 dBZ. Hail size distributions are relatively uncertain but are generally considered to be exponential up to a ceiling diameter (Auer, 1972;Lane et al., 2008). Herein the size distribution of hailstone is assumed to follow (Cheng and English, 1983):

$$N(D) = 115\lambda^{3.63} e^{-\lambda D} \qquad (2)$$

where $D$ is the hailstone diameter (Cheng and English, 1983). This formulation is based on seven events sampled Alberta, Canada which covered a small diameter range than indicated by the RADAR products, but it has the advantage that the distribution requires a single fitting parameter ($\lambda$) and thus can be fully described using only $D_{75}$. As shown by the example hail distribution (expressed in $dN/dR$) for $D_{75} = 25$ mm and $\lambda = 0.053$ mm$^{-1}$ (Fig. 3), the slope of the with hydrometeor diameter is considerably shallower than for rain droplets as described using Marshall-Palmer. In order to avoid the occurrence of extremely large hailstones, we truncate the distribution to include diameters up to two times the RADAR-estimated 75$^{th}$ percentile hail stone diameter ($D_{75}$). The presence of such a hail-size ceiling is consistent with previous observations (Auer, 1972).

Wind speeds from RADAR have been previously used for numerical wind resource verification (Salonen et al., 2011). Wind speeds at a typical wind turbine hub-height of approximately 80 m are derived using the radial wind speeds from the 0.5° elevation angle scan at a distance of 8 km ($\pm$ 0.5 km) range from the RADAR station using an assumption of uniform flow from:

$$V_{radial}(\theta) = V_{mean}\cos(\theta) \qquad (3)$$

where $\theta$ is the difference in angle between the RADAR beam and the direction of mean flow, and $V_{mean}$ is the mean wind speed at hub height. A least-squares fit of a sinusoid of this form is made to each wind speed scan (excluding cells which report a zero wind speed) to estimate $V_{mean}$. The resulting wind speed is then used within the simple description of the blade rotational speed as a function of hub-height wind speed shown in Fig 4c. This operational RPM curve is based on long-term data provided from large operating WT arrays (under an NDA) and represents the mean rotational speed across all WT operating in these arrays as a function of the mean wind speed at hub-height across the arrays. For this reason, the mean RPM begins to decrease at wind speeds below the cut-out velocity (of 25 ms$^{-1}$).

Once the hydrometeor type (rain or hail), hydrometeor size (which determines mass and terminal velocity) and wind speed for a reporting period are known, hydrometeor impact energies for that period are calculated using the mass and closing velocity for hydrometeors of each radius occurring in the period. For this analysis the terminal velocity for each size of rain droplets is derived using (Stull, 2015):

$$V_{t,rain} = k \left[ \frac{\rho_o}{\rho_{air}} R \right]^{1/2} \qquad (4)$$

where $R$ is the droplet radius (m), $k = 220$ m$^{1/2}$s$^{-1}$, $\rho_o$ is air density at sea level (set to a constant of 1.25 kgm$^{-3}$, herein), $\rho_{air}$ is air density at the altitude above sea level at which the rain droplet is crossing the rotor plane (see example of $V_{t,rain}$ in Fig. 3). The terminal velocity of hail stones is derived using (Stull, 2015):





$$V_{t,hail} = \left[\frac{8}{3}\frac{|g|}{C_D}\frac{\rho_i}{\rho_{air}}R\right]^{1/2} \tag{5}$$

where $R$ is radius of the hailstone (m), $\rho_i$ is the density of ice (set to a constant of 900 kgm⁻³ herein), $\rho_{air}$ is air density at the altitude at which the hail is falling. $C_D$=0.55 is the drag coefficient (Stull, 2015) (see example of $V_{t,hail}$ in Fig. 3).

Closing velocity, $V_c$ as a function of hydrometeor type and diameter ($D$) is calculated from wind speed, $V_{mean}$, rotor

5  speed, $V_r$ (calculated from wind speed and RPM curve), terminal velocity, $V_t$ and blade position, $\phi(t)$. $V_r$ as derived here represents the linear speed of the blade tip due to rotation, as this will lead to conservative estimates of impact energy.

$$V_c(D,t) = \left[V_{mean}^2 + (V_r + V_t(D) \cdot \cos(\phi(t)))^2\right]^{1/2} \tag{6}$$

The impact rate ($I$) on the blade leading edge as a function of hydrometeor type and size is calculated from the number density of the hydrometeors of a given diameter ($N(D)$) and the closing velocity:

$$I(D,t) = N(D) \cdot V_c(D,t) \tag{7}$$

The assumption that all falling rain droplets will impact the blade is made on the basis of evidence that only droplets with diameters below 0.2 mm have insufficient inertia to be deflected from the blade by streamline deformation (Eisenberg et al., 2018). The maximum kinetic energy transferred to the blade from the hydrometeors is then computed for each hydrometeor type and diameter using the following approximation:

$$E_K(D,t) = \frac{1}{2}m(D) \cdot V_c(D,t)^2 \tag{8}$$

where $m(D)$ is the mass of the hydrometeors of a given diameter.

The total kinetic energy of impacts over a time interval, $T$, associated with hydrometeors of diameter, $D$, is given by:

$$E_{K,T}(D) = \int_{t_0}^{t_0+T} I(D,t) \cdot E_K(D,t)dt \tag{9}$$

where $dt$ is the time interval at which the RADAR measurements are available (5 minutes).

20  The NEXRAD-estimated probability of hail and the geographic extent of hailfall are both treated probabilistically with respect to the number of expected hail impacts on any particular wind turbine within the wind farm. The number of expected impacts at each kinetic energy are multiplied by two factors representing these two effects. (1) The probability of hail being associated with the storm in question, as estimated by NEXRAD's hail detection algorithm, and (2) the fraction of RADAR cells within 5 km of the wind farm centroid which have a composite reflectivity of > 50 dBZ, the range commonly associated with

25  hail.

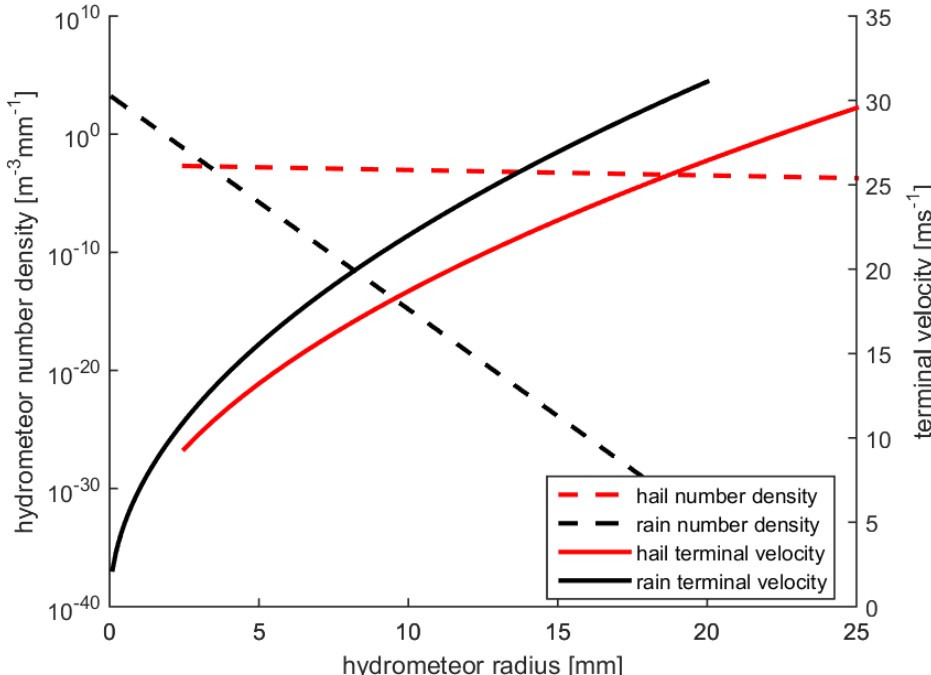

**Figure 3 – Example of the hydrometeor number density (*dN/dR*; number of droplets per meter cubed of air per mm of radius increment) for a precipitation rate of 25 mm hr$^{-1}$ for rain droplets (as described using the Marshall-Palmer size distribution) and for hail stones (for *D$_{75}$* of 25 mm and *λ* = 0.053 mm$^{-1}$) (left axis). Note: The *λ* value employed (*λ* = 0.053 mm$^{-1}$) differs from the range (*λ*: 0.1 to 2 mm$^{-1}$) used**

**by (Cheng and English, 1983) for the seven events they sampled and thus corresponds to a larger maximum hail size. Hydrometeor terminal velocities of hail and rain are shown by radius on the right axis.**

NWS RADAR products have been subject to extensive product development efforts and a wide range of evaluation exercises (Cunha et al., 2015;Villarini and Krajewski, 2010;Straka et al., 2000), but are nevertheless associated with measurement uncertainties, as are the approximations applied herein to derive terminal fall velocities and kinetic energy transfer.

To provide a first assessment of how these uncertainties in input data propagate through the analysis framework and thus impact derived kinetic energy exchange each of three key parameters of the erosion potential are perturbed from 50% to 150% of observed values during two example periods of comparatively high erosion potential. The first case represents a period of large hail. In this analysis $D_{75}$ is set to the 99[th] percentile $D_{75}$ at KMKX (42 mm) and $V_{mean}$ is set to 11.3 ms$^{-1}$ (i.e. the mean wind speed conditionally sampled by ±10% of $D_{75}$ = 42 mm). In the second, a heavy rain event is considered. The *RR* is set to the 99[th]

percentile value at KMKX (18 mmhr$^{-1}$) and the $V_{mean}$ is set to the mean value (12.8 ms$^{-1}$) during heavy rainfall (i.e. *RR* within 10% of the 99[th] percentile value at KMKX).

Uncertainties in RADAR-derived hail sizes are less well characterized than for $V_{mean}$ and *RR*. For *RR* the range of ± 50% is inclusive of previously published uncertainties, understanding that those uncertainties are a function of spatial resolution, *RR* and RADAR processing algorithm (Seo and Krajewski, 2010;Seo et al., 2015). Wind speed uncertainty (as quantified using RMSE)

for an elevation angle of 0.5° is approximately ± 3.4 ms$^{-1}$ (Fast et al., 2008) and thus for a wind speed of 12.8 ms$^{-1}$ a +/-50% variation is fully inclusive of the estimated wind speed error.



## 3    Results

Key aspects of the erosion-relevant RADAR-derived atmospheric properties at the six locations are summarized in Fig. 4. Consistent with previous precipitation climatologies, there are marked spatial gradients in the annual total and precipitation intensity (*RR*, Fig. 4a) (Prat and Nelson, 2015). Precipitation rates of $< 5$ mmhr$^{-1}$ are common at all sites, *RR* of 20 mmhr$^{-1}$ are

experienced at all locations, but only the site in Texas (KMAF) exhibits any occurrence of rainfall intensity in excess of 35 mmhr$^{-1}$. Using a damage rate of $3\times10^{-5}$ s$^{-1}$ for a *RR* of 20 mmhr$^{-1}$ and a closing velocity of 120 ms$^{-1}$ (Eisenberg et al., 2018), the frequency of *RR* of 20 mmhr$^{-1}$ at the site in Texas is such that it would accumulate ~ 0.6 of impact necessary to reach the transition threshold from the incubation region to material loss over a 25 year period.

At most sites, snow and ice occur at rates at least one order of magnitude less frequently than rain. The exception is the site

in Idaho (KSFX) (Fig. 4d). At each of the six locations, there are fewer than 40 five-minute hail periods per year. Consistent with expectations and previous research (Cintineo et al., 2012), while hail events occur at all six sites, hail frequency and severe hail events (with maximum hail sizes $> 25$ mm) are substantially more frequent at the nominal wind farm locations in Texas, Illinois and Wisconsin (RADAR ID; KMAF, KDVN and KMKX) (Fig. 4b). The derived frequency distributions of wind speed close to wind turbine hub-heights (WT HH) exhibit a high frequency of wind speeds above typical wind turbine cut-in speeds, and are

particularly right-skewed at the site in Illinois (Fig. 4c). The mean annual wind speed near nominal WT HH is lowest at KSFX (in Idaho) where it is $\approx 5.9$ ms$^{-1}$. They range from 8.6 to 10 ms$^{-1}$ at KGRR, KMKX, KBUF and KDVN (listed in ascending order of $V_{mean}$). The wind speed distributions at these five of the six locations exhibit relatively good qualitative agreement with *a priori* expectations (see wind resource maps available at; https://windexchange.energy.gov/maps-data/324) and estimates from simulations for 2002-2016 with the Weather Research and Forecasting model (Pryor et al., 2018) for 12 km grid cells containing

the nominal wind field locations that indicate mean annual wind speeds of 6.5 ms$^{-1}$ at KSFX, and 8.4-9.0 ms$^{-1}$ (KGRR, KMKK, KBUF and KDVN). However, wind speeds derived from RADAR observations from KMAF are relatively low (mean value of 5.9 ms$^{-1}$) and exhibit a relatively low frequency of observations above 13 ms$^{-1}$ (2.2%). This negative bias (of $> 1$ ms$^{-1}$ in the mean relative to the resource map and WRF model output) from the Texas site will tend to lead to lower RPM values and hence blade tip speeds and thus a negative bias in kinetic energy transfer at this location. Wind speed distributions during precipitation

and no-precipitation periods are qualitatively similar at all six locations. Modal values are within +/- 1.2 ms$^{-1}$, but the distributions are heavier tailed at all sites during precipitation periods. Mean wind speeds during precipitation are 0.2-3.8 ms$^{-1}$ higher at the six locations than during times of no precipitation.



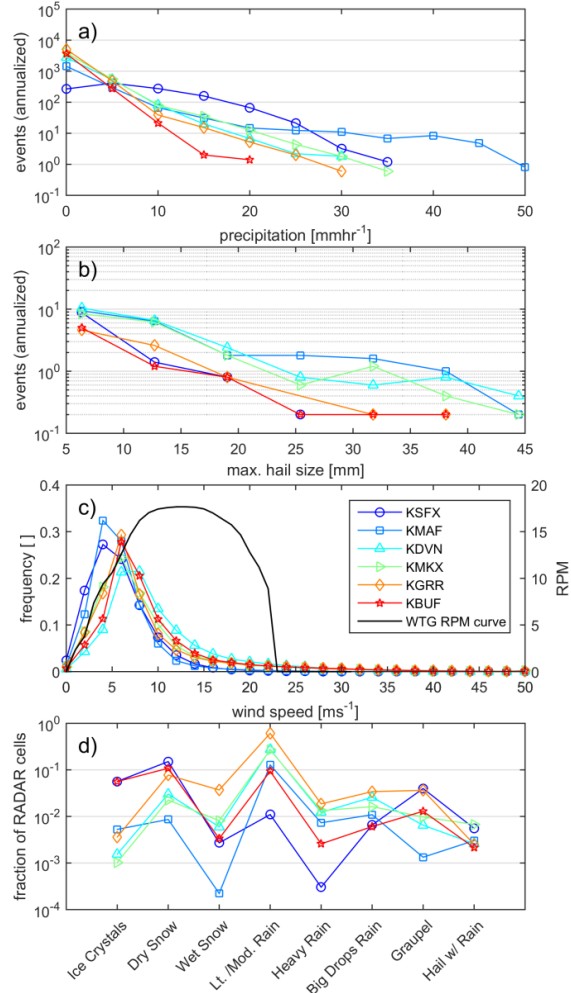

**Figure 4 – Precipitation and wind speed climates from RADAR data (see locations in Fig. 1). (a) Mean annual number of 5 minute periods of each *RR* intensity class (discretized in 5 mmhr⁻¹ intervals). (b) Mean annual number of 5 minute periods with maximum hail sizes (*D₇₅*) (discretized in 5 mm intervals). (c) Wind speed distributions for all 5-minute periods (discretized in 2ms⁻¹ intervals). The black line**

**in this frame shows the WT RPM curve as a function of wind speed (WTG RPM, right axis) (d). Occurrence of NWS RADAR hydrometeor classifications for each nominal wind farm shown as the fraction of RADAR cells in each class during all periods with *RR* > 1 mmh⁻¹.**

Given the exponential dependence of hailstone and rain droplet size on precipitation intensity and the accumulated

damage therefrom (Eisenberg et al., 2018), the distributions of kinetic energy transfer from the two hydrometeor types at all sites are heavy-tailed. Further, the probability distributions of each 5-minute estimate of kinetic energy transfer (Fig. 5) and total annual kinetic energy transfer (Fig. 6) indicate marked differences between the sites and between the two hydrometeor types. Extremely high hail kinetic energies are most frequently projected for sites in Texas (KMAF), Illinois (KDVN) and Wisconsin (KMKX) (Fig 5a). This is consistent with the precipitation climatology summarized in Fig. 4b and the high frequency of wind

speeds associated with high WT RPM (Fig. 4c). At these three sites some events (5 minute periods) exhibit kinetic energy of



transfer from hail in excess of 300 J (Appleby-Thomas et al., 2011;Kim and Kedward, 2000) (Fig. 5a). Although these events have a low probability (less than 1 per square meter per year), they may thus be sufficient to cause damage to blade coatings in isolation from the effects of the cumulative fatigue. Conversely, individual rain impacts rarely exceed 5.2 J at any site. The probability of exceeding this impact kinetic energy threshold over a square meter of blade leading edge is less that $10^{-3}$ per year

5 (Fig. 5b). Thus, hail dominates the annualized cumulative kinetic energy of transfer to each square meter of the blades at all sites (Fig. 6). Indeed, at all sites, despite the low probability of hail relative to rain (cf. Fig. 4a and 4b), total annual kinetic energy transfer from hail exceeds that from rain by at least two orders of magnitude (Fig. 6). The lowest cumulative kinetic energy transfer is projected for the nominal wind farm sites in Idaho (KSFX), New York state (KBUF) and in Michigan (KGRR). Conversely, values are highest for in Texas (KMAF), Illinois (KDVN) and Wisconsin (KMKX). This is consistent with previous

10  characterizations of hail frequency, which show hailfall to be most common in the Great Plains, and very infrequent west of 105° west (Fig. 1) (Cintineo et al., 2012;Allen and Tippett, 2015).

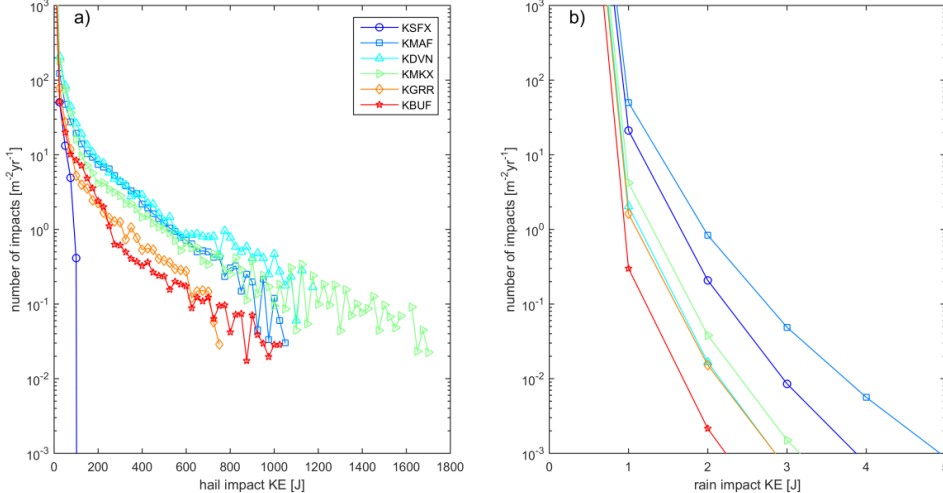

**Figure 5 – Histograms of kinetic energy of hydrometeor impacts. Annual number of (a) hail and (b) rain impacts per m² of blade leading edge as a function of impact kinetic energy. The y-axis in panel (b) has been truncated to a maximum value of 1000 per year.**

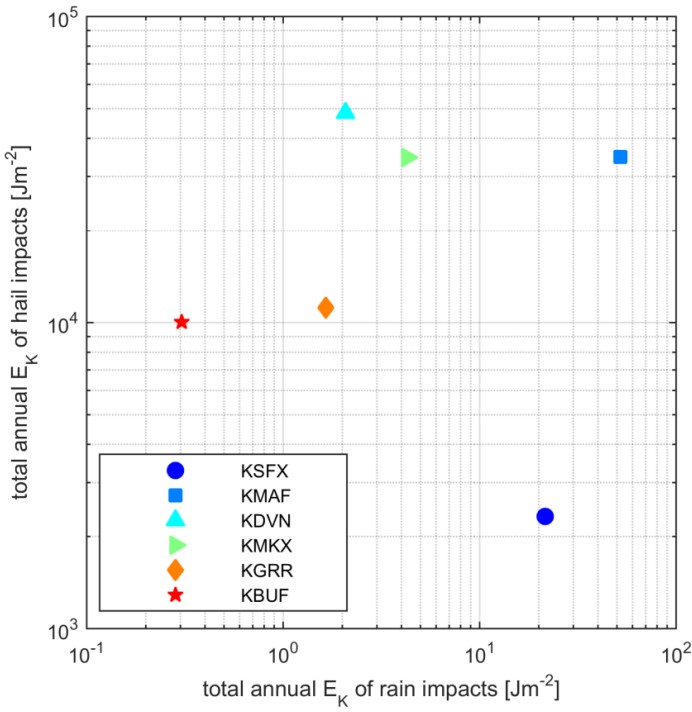

**Figure 6 – Total annual kinetic energy ($E_K$) per m$^2$ of blade leading edge from rain and hail impacts at each location.**

Fig. 7 illustrates that only a very small fraction of 5-minute periods dominate kinetic energy transfer to the blades from both hail and rain. At all sites over 80% of rain-induced kinetic energy transfer occurs in the top 80 5-minute periods per year. Indeed, at all but the site in Idaho (KFSX) over half of the total rain-induced kinetic energy transfer to the blade occurs in only 20 5-minute periods in a year. The probability distribution of hail-induced kinetic energy transfer is even more heavy-tailed with at all sites 90% of the cumulative kinetic energy transfer to the blades from hail occurring in few than 25 5-minute periods per year. Thus, few events dominate the annual total accumulated impact damage.

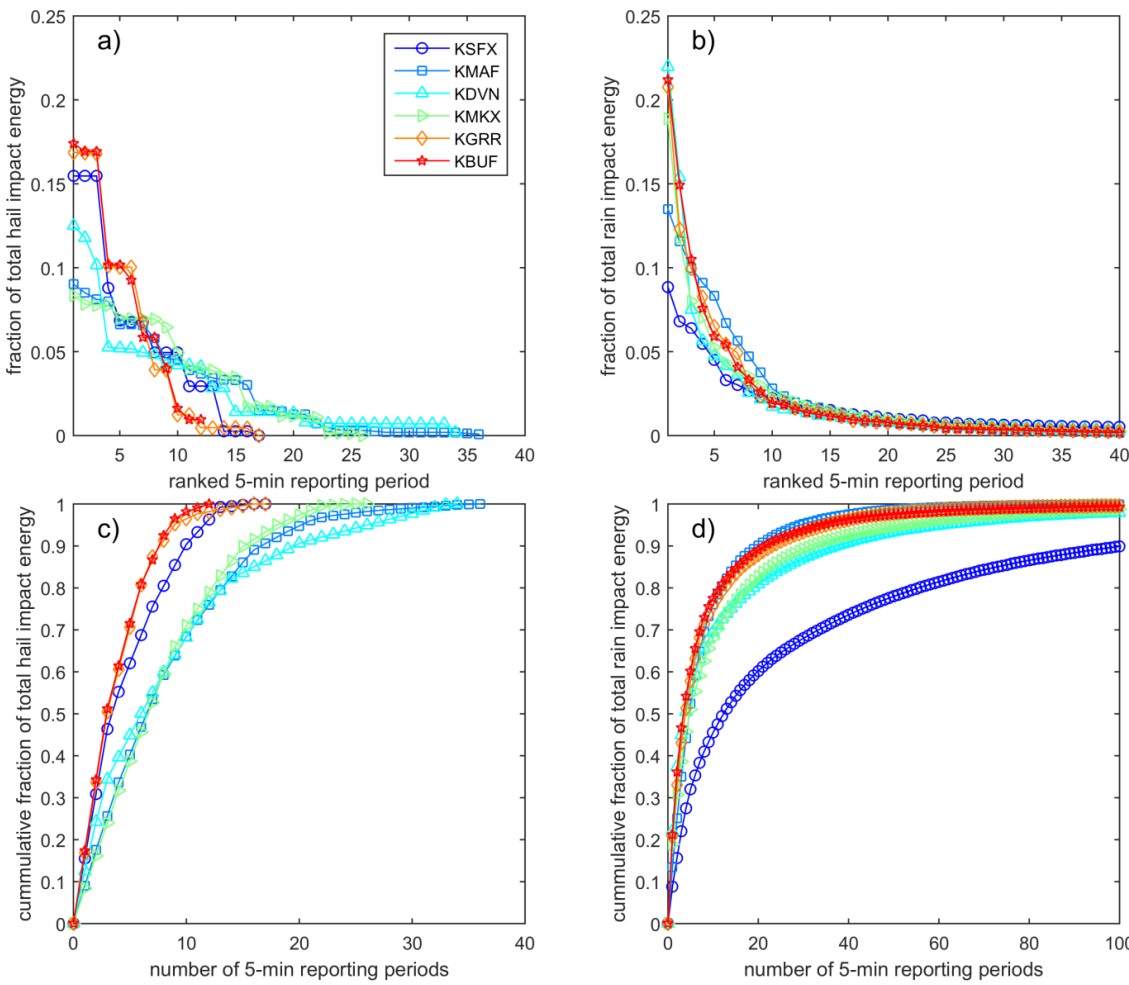

**Figure 7 – Contributions of the most intense precipitation events to annual total kinetic energy from hydrometeor impacts. (a) Contribution of the top 40 5-minute periods of hail as a fraction of the annual total kinetic energy of hail impacts (b) Contribution of the top 40 5-minute periods of rain as a fraction of the annual total kinetic energy of rain impacts. Cumulative fraction of annual impact kinetic energy from the top X (c) hail events and (d) rain events, where X is set to 40 for hail because no site exhibits more than 36 events per year and is truncated to 100 for rain.**

Illustrative examples of uncertainties in impact kinetic energy due to NEXRAD observational uncertainties in $V_{mean}$, $RR$ and $D_{75}$ are shown in Fig. 8. For the representative 5-minute period of heavy rain variation of $RR \pm 50\%$, is associated with a $\pm$ 15% variation in kinetic energy of impact (Fig. 8a). Increases or decreases in mean wind speed by 4.2 ms$^{-1}$, (the upper end of wind speed uncertainty observed in previous work for an elevation angle of 0.5°) (Fast et al., 2008), are shown to decrease kinetic energy, since rotor speed decreases for wind speeds below, or significantly above the rated wind speed of the turbines (Fig. 4c). For a representative period of hail ($D_{75}$ = 42 mm and $V_{mean}$ = 11.3 ms$^{-1}$), impact kinetic energy varies by $\pm$ 20% for a $\pm$





50% variation in $V_{mean}$ and $D_{75}$ (Fig. 8b). Impact kinetic energy actually decreases as $D_{75}$ exceeds 120% of the nominal value ($D_{75}$ = 42 mm) (Fig. 8b). This decrease is explained by the interaction of the single parameter exponential hail size distribution (Fig. 4c) and the applied hail diameter ceiling. As $D_{75}$ increases the truncation of the upper tail of the hail distribution (Fig. 8c) means the total modelled mass of hail per unit volume decreases (Fig. 8d).

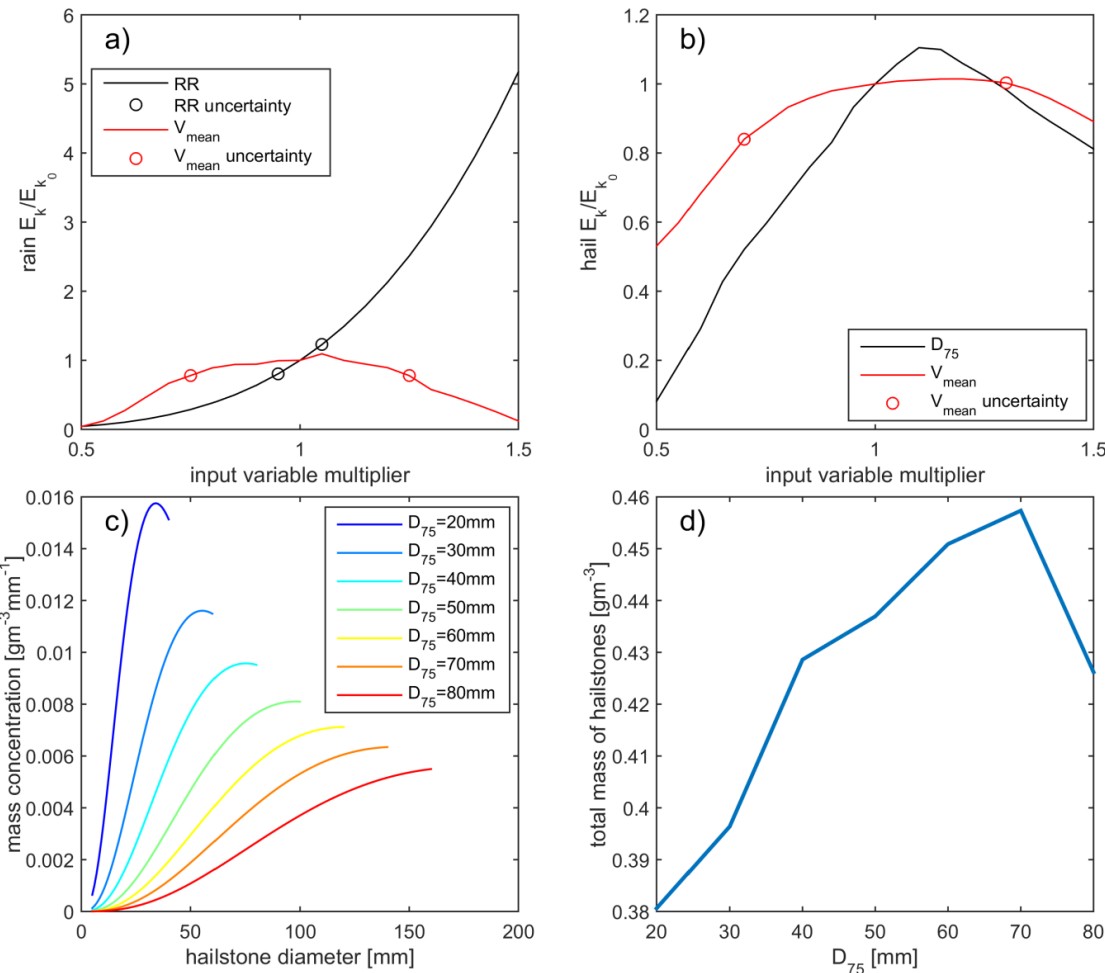

Figure 8 – Sensitivities of rain (a) and hail (b) impact kinetic energies in one 5-minute period to application of ±50% uncertainties on the input parameters; wind speed ($V_{mean}$) and precipitation intensity ($RR$) or hail diameter ($D_{75}$). Circles represent reported uncertainties in RADAR retrievals of wind speed (Fast et al., 2008) and rainfall rate (see Table 1 in (Seo and Krajewski, 2010)). (c) Mass concentrations

10    of hailstones per cubic meter of air (expressed as $dM/dD$) associated with a range of $D_{75}$ values as a function of hailstone diameter. (d) Total hail mass (in g) per m³ of air as a function of $D_{75}$.



## 4    Conclusions

A robust and flexible framework has been developed and presented for generating an observationally constrained georeferenced assessment of precipitation-induced wind turbine blade leading edge erosion potential. The approach elaborated herein is naturally subject to a range of uncertainties but is automated, objective, repeatable and predicated on publicly available

data available from across most of the continental US. Further, the modular structure means it is flexible to use of different assumptions and/or data streams. Although the data volumes are not trivial (see Appendix A) this analysis framework could be applied to NWS RADAR data to estimate LEE potential at any arbitrary site in CONUS and/or applied to data from other national dual-polarization RADAR networks for other regions of the world. The tool proposed here could be used to provide a first assessment of the erosion climate in which a given sited turbine may operate in. It thus provides an important first step

towards enabling an assessment of the threat of excessive precipitation-induced LEE in a given deployment environment and the cost-effectiveness of options to reduce the likelihood of premature blade damage.

The actual likelihood of excess WT LEE and blade damage in any environment is naturally not only a function of the precipitation and wind climate, but also the WT dimensions, materials used in the blade coatings and the coating thickness (Eisenberg et al., 2018;Slot et al., 2015), the presence of existing micro-structural defects (Evans et al., 1980) due to

manufacturing defects and damage during transportation (Keegan et al., 2013;Nelson et al., 2017) and other aspects of the operating environment (including thermal fatigue and the occurrence of icing (Slot et al., 2015)).

The preliminary estimates of erosion potential and the partitioning between liquid precipitation and hail are naturally subject to limitations including:

- The relatively short duration of time for which the dual-polarization RADAR products are available. The upgrade of the

NWS RADAR network to dual polarization was completed in April 2013, thus only the complete years of 2014-2018, inclusive were available for analysis. Given the large inter-annual variability in precipitation climates this is too short to build a comprehensive climatology (Karl et al., 1995;Prein and Holland, 2018). Any geospatial depiction of the potential precipitation erosion climate will vary according to the precise data period used to compute the climatology and may evolve as a result of climate non-stationarity altering aspects of the precipitation climate (e.g. probability of hail (Brimelow et al.,

2017) and rainfall intensity (Easterling et al., 2000)).

- Assumptions applied in deriving precipitation intensity and other precipitation properties from RADAR. Notable event-to-event variations in the applicability of Z-R relationships have been reported during rain (Uijlenhoet, 2001;Villarini and Krajewski, 2010).

- Assumptions regarding the size distribution of rain droplets. Most observational studies indicate an exponential form

(Uijlenhoet, 2001), and the Marshall-Palmer distribution is the most widely applied. However, a range of different forms have been proposed to describe the size spectrum of rain droplets ($dN/dR$) including gamma (Ulbrich, 1983) and lognormal (Feingold and Levin, 1986), an alternative exponential form (Best, 1950) and more complex non-parametric forms (Morrison et al., 2019). There is also evidence that droplet size distributions may exhibit a functional dependence on near-surface wind speed (Testik and Pei, 2017).

- Assumptions regarding the size distribution and occurrence of hail (Dessens et al., 2015;Allen et al., 2017). The evolution of the NWS RADAR network to dual polarization provides an unprecedented opportunity for spatial estimates of hail occurrence and size in clouds (Kumjian et al., 2018). However, hail production is a complex and incompletely understood phenomenon (Dennis and Kumjian, 2017;Blair et al., 2017;Pruppacher and Klett, 2010) and there are substantial event-to-event variations in the size distribution of hail stones and in the presence of solid-phase hydrometeors in clouds (as detected





by RADAR) and the occurrence of hail at the ground (Kumjian et al., 2019). Estimates of hail occurrence and size distribution presented herein are likely conservative with respect to both properties (i.e. both the frequency and size of hail stones may be over-estimated).

- The applicability of the RADAR-derived wind speed estimates to derive wind turbine blade rotational speed. There are considerable challenges to line-of-sight wind retrievals from RADAR (Fast et al., 2008). The approach adopted herein assumes a uniform wind flow pattern to derive the wind speeds at the nominal wind turbine hub-height which may not be realized. As described herein while the wind speed climates at five of the six locations considered herein exhibited relatively good agreement with previous estimates of wind climates, values for the location in Texas are negatively biased. This likely results in a negative bias in kinetic energy transfer for this site.

Future work could address and reduce these uncertainties and adapt this approach to examine different wind turbines (by applying a different RPM curve) and/or to assimilate different atmospheric data and/or incorporate more explicit aspects of materials response. In this analysis we have chosen to focus on an energetic approach in which we compute the accumulated kinetic energy transmitted to the blade leading edge instead of using approaches based on the waterhammer equation that seek to compute the impact pressure and material response to the resulting Rayleigh, shear and compression waves (that are assumed to act independently from each individual impact) (Slot et al., 2015;Dashtkar et al., 2019). It is important to reiterate that the approach adopted here i.e. to compute the maximum total kinetic energy transferred to the blade, which is used here as a proxy for the erosion potential, represents the upper bound on actual kinetic energy transfer since it assumes all falling hydrometeors impact the blade, and neglects energy loss during the transfer, 'splash' and bounce of hydrometeors. There are more complex frameworks that can be applied to simulate the pressure and transient stresses on the blade coatings (Mishnaevsky Jr, 2019) and impingement erosion (Amirzadeh et al., 2017b, a). A model of the blade response to precipitation impacts could be incorporated within the analysis framework to examine the probability and time to exceed the (Cumulative) Failure Threshold Energy (Fiore et al., 2015).

This work suggests the dominance of hail as a damage vector for WT blades at all of the sites studied here. This finding indicates the key importance of efforts to build and enhance hail climatologies (Allen et al., 2015;Gagne et al., 2019) with applications in a wide range of industries (from insurance to renewable energy). The dominance of hail as a damage vector and the importance of a relatively small number of 5-minute periods to total annual kinetic energy transfer from rain adds credence to the proposal that blade LEE could be greatly reduced by operating erosion-safe turbine control (Bech et al., 2018) wherein the WT are curtailed during periods with extreme precipitation (very heavy rain or the occurrence of hail) without substantial loss of income.

## 5 Acknowledgments

This research was funded by the US Department of Energy (DE-SC0016438) and Cornell University's Atkinson Center for a Sustainable Future (ACSF-sp2279-2018). It was enabled by access to computational resources supported via the NSF Extreme Science and Engineering Discovery Environment (XSEDE) (award TG-ATM170024) and ACI-1541215. The authors gratefully acknowledge the scientists and technicians of the National Weather Service for their work in realizing the dual polarization RADAR network and making the data publicly available.



**Data availability**

The USGS Wind Turbine Database used in Figure 1 is available for download from https://eerscmap.usgs.gov/uswtdb/. The NOAA Weather and Climate Toolkit (WCT) is a free, platform independent Java-based software tool distributed from NOAA's National Centers for Environmental Information (NCEI) (download is available from https://www.ncdc.noaa.gov/wct/). The

NWS RADAR data are available from download from; https://www.ncdc.noaa.gov/data-access/radar-data.

**Appendix A.**

The workflow, NWS RADAR data products and data volumes necessary for the components of the precipitation erosion climate are as follows:

1. Download daily .tar archives of NEXRAD polarized Doppler RADAR data using ftp from the data repository hosted at;
https://www.ncdc.noaa.gov/nexradinv/. These tar archives contain at 5 minute intervals all NEXRAD level 2 and 3 data and data products, in binary NEXRAD format. The 365 daily tar comprise 60 – 100 GB per station per year (PSPY).
2. Preprocessing
   a. Extract hourly precipitation (N1P) and hail reports (NHI) files from each daily tar file.
   b. Import raw files into NOAA Weather and Climate Toolkit (https://www.ncdc.noaa.gov/wct/), translate N1P, N0V and
NHI files into netcdf, and .csv, File sizes and numbers
   - Hydrometeor Classification (HHC) raw files (32,000 to 70,000 files PSPY, 124 – 180 MB PSPY)
   - NHI csv files (12,000 to 137,000 files PSPY, totaling 25 – 190 MB PSPY)
   - N1P raw files (68,000 to 90,000 files PSPY, totaling 600 – 900 MB PSPY)
   - N1P netcdf files (130,000 to 150,000 files PSPY, totaling 240 – 290 GB PSPY)
- Base wind speed (N0V) raw files (40,000 to 60,000 files PSPY, totaling 35-45 GB PSPY)
   - N0V netcdf files (40,000 to 60,000 files PSPY, totaling 900 – 1100 MB PSPY)

Subsequent data analysis is conducted within MATLAB.

**Author contributions**

SCP and RJB jointly designed the project and obtained the funding and computing resources for the project. FL conducted the majority of the data analysis and developed the figures with input from SCP and RJB. All contributed to writing the paper.

**Competing interests**

The authors declare they have no conflict of interest.

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
