# Peer review of "RADAR-Derived Precipitation Climatology for Wind Turbine Blade Leading Edge Erosion"

_Wind Energy Science, 2019_

## Referee Comment (RC1) · Anonymous Referee #1 · 2 Oct 2019

**Review for *Wind Energy Science* of**

**RADAR-Derived Precipitation Climatology for Wind Turbine Blade Leading Edge Erosion**

by F. Letson, R.J. Barthelmie, and S.C. Pryor

**General Comments**

This paper computes the energy of impact to wind turbines by rain and hail, a topic that is seldom discussed, yet very important to operation and maintenance of wind farms. It points out the potential for these hydrometeors to erode the leading edge of wind turbines. They leverage public data for the US regarding dual-pol radar data and estimate the energy for typical turbine parameters. This paper should be of interest to readers of WES. The work is well-justified, interesting, and well presented. The authors justify that the problem is important and review the subject appropriately. The calculations are appropriate, but quite conservative, as the tip speed is used in computing the energy of impact. In fact, the tip itself would be seldom hit. It may be useful to provide a range of impacts as a function of distance from the hub.

**Specific Comments**

- The literature review is appropriate, but one wonders how much similar research has been done by the insurance industry regarding damage to cars and roofs.
- This paper focuses on the U.S. It would be helpful to discuss how it is expected to apply in other parts of the world.
- P. 3, line 11 – reference to the work of Bech et al. for Denmark on curtailment is interesting. Did they do a cost-benefit analysis? Readers may be interested.
- P. 5, line 5 – It would be interesting to comment on the accuracy of the current classification methods of the dual pol radars – it used to be rather poor, but may have improved with time.
- P. 6, line 7 – Please clarify what you mean by the "largest values". Do you mean the 75% from the prior page? Later on line 15, you refer to a fitting parameter for $D\_75$, which suggests that.
- P. 6, line 29 – "the mean RPM begins to decrease at wind speeds below the cut-out velocity …" Is this true? Please provide references. The power remains constant at rated capacity until nearly at cut-out, so how does the RPM decrease? Or do you mean as it approaches cut-out speed? Please clarify
- P. 7, line 6 – The rest of the analysis uses the speed of the blade tip for calculations. What if you used the mid-point of the blade instead as a more representative speed? You do mention that this is "conservative". It would be informative to compare the speed at the mid-point to help show the variability across the blade. Or even to plot impact as a function of distance from the hub.
- P. 15 – lines 19-p. 16, line 10 - -This is a nice list of limitations. Thanks for providing.

**Technical corrections**

- P. 1, line 26 – please write out "approximately" (not approx..)
- Throughout – please add space between references in parentheses
- Throughout – it is difficult to follow and remember the 4-letter designations for the radar sites, even for an American reviewer. It will likely be even more difficult for international readers. Perhaps using the states where they are located in your references to the sites would enhance readability??
- P. 2, line 30 – WT was already defined
- P. 2, line 35 – please specify U.S. Central Plains – this in an international journal.
- P. 3, line 6 – please define CONUS first time it is used.
- P. 4, line 9 – "nominal wind farm located within the observation areas of six RADARS" is confusing.  Are you referring to a wind farm for each of the six RADARS? What do you mean by "nominal" wind farm?  Is this "notional"?  Have you identified a specific wind farm or are you referring to one within the reach of the radar beam? As written, it implies that you have identified a wind farm that is reached by the beams of all 6 radars, which is surely not what you meant to say.
- P. 5, line 17 – I don't think you mean to refer to Fig. 3 here.
- P. 6, line 11 – "hailstones" should be plural
- P. 6, line 13 – "sampled IN Alberta"
- P. 6, line 27 – "wind speed AS shown in …"
- P. 11, line 10 – hail storms are quite frequent in Boulder, Co which is just west of 105°
- P. 12, line 7 – occurring in fewer that …" (not few).  The point you make here is certainly true for damage to roofs and cars.
- P. 15, lin 5 – " it is flexible to use with different …" (not "of")
- P. 15, line 12 – please indent paragraphs.
- P. 16, lines 5-7 – The word "herein" is used 3 times in 3 lines.

---

## Referee Comment (RC2) · Anonymous Referee #2 · 17 Oct 2019

The paper by Letson et al. presents an interesting study on an emerging research field related to wind energy. It is of high value for readers of WES. The methodology is presented in a clear way. The results are discussed at a high level with many papers cited within the cross-cutting field. The results are novel. The conclusion summarizes in outstanding way the many learnings and perspectives for further research. The paper is recommended for publication.

Minor edits

P.1, line 26. Would the repair cost be different between on- and offshore wind farms? Please clarify.

P. 2, line 38. Another challenge than age alone could be US East Coast wind farms.

Please elaborate.

P. 3, Figure 1. As also suggested by Reviewer 1, the naming of radar stations is confusing. The naming could for simplicity be the state names (ID, TX, IL, WI, MI and NY) as you selected only one radar station in each state. For clarity, the full station names could be given once. You have selected a wind farm within a radius of roughly 60 km from each radar station. Would it be possible to add circles of this dimension at each station in the map?

P. 5, line 1. Hourly precipitation rate (N1P). Did you average to 1 hour or is the data in the database 1-hour data?

P. 5, line 8. Is the NCR maybe Normalized Composite Reflectivity?

P. 5, line 20. Is the spatial mean based on 5-minute and 6-minute values, then averaged to hourly?

P. 6, line 4, m-4. Do not use italic.

P. 6, line 14, would smaller be more correct than small?

P.6, line 16, . . .the slope of the with hydrometeor diameter.. I wonder if "with" is a good formulation here

P. 6, line 16. . .is considerably shallower than. . .. Would it be more clear to say nearly constant?

Figures 3 to 7. The graphics are good but can be improved. Please use capital letter at the legends at all axis.

Fig. 4, upper panel. You show values at 0 mm/hour. Is this indicating all events with no precipitation (or from >0 but smaller than 5 mm/hour). Please clarify. Add simplified station names.

Fig. 5 and Fig. 7. Add simplified station names.

Fig. 6, Add simplified stations names near each of the colored markers, if possible.

Page 9. The discussion of the mean wind speed for all cases at six stations from three information sources and in addition wind speed for rainy cases for six stations from one source. It is a bit confusing. A table summarizing the relevant numbers could be easier for readers to follow the discussion.

P.6 line 7 and P. 16, line 2 you write 'conservative'. I am in doubt what you mean in this context. Please clarify.

---

## Editor Comment (EC1) · Julio J. Melero (Editor) · 17 Oct 2019

Dear authors, please post your response to the reviewers comments. I kindly ask you to address the point recommendations given by both reviewers in order to let me evaluate in detail the modifications that you incorporate in the final paper.

---

## Author Comment (AC1) · 8 Nov 2019

**REPSONSE TO REVIEWERS**

RADAR-Derived Precipitation Climatology for Wind Turbine Blade Leading Edge Erosion

**F. Letson, R.J. Barthelmie, S.C. Pryor**

MS No.: wes-2019-43

**MS Type: Research article**

**Special Issue: Wind Energy Science Conference 2019**

**Julio J. Melero (Editor) melero@unizar.es Received and published: 17 October 2019**

Dear authors, please post your response to the reviewers comments.

I kindly ask you to address the point recommendations given by both reviewers in order to let me evaluate in detail the modifications that you incorporate in the final paper.

The authors would like to thank the two anonymous reviewers for their helpful comments. Our responses are enumerated below in bold, aligned to the right side of the page. In cases where the line number of an edit has changed from the previous version of the manuscript, the new line number is included [in square brackets]. A tracked-changes version of the manuscript is included, below.

**Anonymous Referee #1**

**General Comments**

This paper computes the energy of impact to wind turbines by rain and hail, a topic that is seldom discussed, yet very important to operation and maintenance of wind farms. It points out the potential for these hydrometeors to erode the leading edge of wind turbines. They leverage public data for the US regarding dual-pol radar data and estimate the energy for typical turbine parameters. This paper should be of interest to readers of WES. The work is well-justified, interesting, and well presented. The authors justify that the problem is important and review the subject appropriately. The calculations are appropriate, but quite conservative, as the tip speed is used in computing the energy of impact. In fact, the tip itself would be seldom hit. It may be useful to provide a range of impacts as a function of distance from the hub.

Specific Comments (page and line numbers refers to tracked changes version of the manuscript)

1. The literature review is appropriate, but one wonders how much similar research has been done by the insurance industry regarding damage to cars and roofs.

This is an excellent point. New text (and references) has been added to the introduction that documents economic losses for the USA [Page 1 Lines 27 to Page 2 Line 5]

2. This paper focuses on the U.S. It would be helpful to discuss how it is expected to apply in other parts of the world.

The extent of RADAR coverage and the frequency of hail are both reasons why a study like this one makes particular sense in the US. We have added a sentence (and a reference) about the comparatively high frequency of hail events in the central US [Page 2 Lines 1-3] and information about broader applicability (e.g. to the European RADAR network to the conclusions [Page 17-18]

 P. 3, line 11 – reference to the work of Bech et al. for Denmark on curtailment is interesting. Did they do a cost-benefit analysis? Readers may be interested.
 Yos, they did. Bech et al. found their curtailment strategy to be cost effective. This has been

**Yes, they did. Bech et al. found their curtailment strategy to be cost effective. This has been added to the introduction [page 3 line 26-30]**

- P. 5, line 5 It would be interesting to comment on the accuracy of the current classification methods of the dual pol radars – it used to be rather poor, but may have improved with time. Good point. A sentence on improvements in classification has been added. [Page 5 line 25-27]
- 5. P. 6, line 7 Please clarify what you mean by the "largest values". Do you mean the 75% from the prior page? Later on line 15, you refer to a fitting parameter for D\_75, which suggests that.

**Yes. Thank you. This has been made explicit. [Page 6 Line 17-19]**

6. P. 6, line 29 – "the mean RPM begins to decrease at wind speeds below the cut-out velocity …" Is this true? Please provide references. The power remains constant at rated capacity until nearly at cut-out, so how does the RPM decrease? Or do you mean as it approaches cut-out speed? Please clarify

**This is based on our analysis of SCADA data. There are a significant number of belowrated-RPM events for wind speeds approaching cutout. This passage has been re-worded to clarify. [Page 7 Line 21-22]**

7. P. 7, line 6 – The rest of the analysis uses the speed of the blade tip for calculations. What if you used the mid-point of the blade instead as a more representative speed? You do mention that this is "conservative". It would be informative to compare the speed at the mid-point to help show the variability across the blade. Or even to plot impact as a function of distance from the hub.

**Leading edge erosion is primarily of concern at or near the blade tip. An explanation (including the fact that local blade speeds vary linearly with distance from the hub) and reference have been added. [Page 7 Lines 36-38]**

8. P. 15 – lines 19-p. 16, line 10 - -This is a nice list of limitations. Thanks for providing. **Thank you.**

**Technical corrections**

9. P. 1, line 26 – please write out "approximately" (not approx..)

Done. Thank you.

10. Throughout – please add space between references in parentheses

Done

11. Throughout – it is difficult to follow and remember the 4-letter designations for the radar sites, even for an American reviewer. It will likely be even more difficult for international readers. Perhaps using the states where they are located in your references to the sites would enhance readability??

**Good idea. We have added state abbreviations to the RADAR station codes throughout the text and figures.**

- 12. P. 2, line 30 WT was already defined
  Thank you 'WT' is now used here without the redundant definition [Page 3 Line 3]
  13. P. 2, line 35 please specify U.S. Central Plains this in an international journal.
- Thank you, this has been added [Page 3 lines 9-10]
- 14. P. 3, line 6 please define CONUS first time it is used.

**Done [Page 1 Line 29-30].**

15. P. 4, line 9 – "nominal wind farm located within the observation areas of six RADARS" is confusing. Are you referring to a wind farm for each of the six RADARS? What do you mean by "nominal" wind farm? Is this "notional"? Have you identified a specific wind farm or are you referring to one within the reach of the radar beam? As written, it implies that you have identified a wind farm that is reached by the beams of all 6 radars, which is surely not what you meant to say.

**Thank you, the wording has been changed to make it clear that there 6 actual wind farms, each covered by a RADAR station [Page 4 lines 12-15]**

- 16. P. 5, line 17 I don't think you mean to refer to Fig. 3 here. **This reference to Fig 3 has been removed [Page 6 Line 7]**
- 17. P. 6, line 11 "hailstones" should be plural
- This has been corrected [Page 7 line 1]

18. P. 6, line 13 – "sampled IN Alberta"

- Yes, thank you. Done. [Page 7 Line 4]
- 19. P. 6, line 27 "wind speed AS shown in ..."

- Done. [Page 7 Line 18]
- 20. P. 11, line 10 hail storms are quite frequent in Boulder, Co which is just west of 105°
   'Very infrequent' has been softened to 'much less frequent', which is still consistent with the literature cited (Cintineo et al and Allen and Tippett) [Page 12, Line 20]
   21. P. 40, line 7.
- 21. P. 12, line 7 occurring in fewer that ..." (not few). The point you make here is certainly true for damage to roofs and cars.

Thank you. This has been corrected [Page 14, line 5]

22. P. 15, line 5 – " it is flexible to use with different ..." (not "of")

Corrected. [Page 17, line 2]

23. P. 15, line 12 – please indent paragraphs.

Done. Thank you. [Page 17, Line 11]

24. P. 16, lines 5-7 – The word "herein" is used 3 times in 3 lines.
 The wording has been changed to avoid this repetition. Thank you. [Page 17, Lines 26 to 29]

**Anonymous Referee #2**

The paper by Letson et al. presents an interesting study on an emerging research field related to wind energy. It is of high value for readers of WES. The methodology is presented in a clear way. The results are discussed at a high level with many papers cited within the cross-cutting field. The results are novel. The conclusion summarizes in outstanding way the many learnings and perspectives for further research. The paper is recommended for publication.

**Minor edits (page and line numbers refers to tracked changes version of the manuscript)**

1. P.1, line 26. Would the repair cost be different between on- and offshore wind farms? Please clarify.

**Text (and a reference) has been added about the higher cost of maintenance for offshore turbines [Page 1 lines 27-28]**

2. P. 2, line 38. Another challenge than age alone could be US East Coast wind farms. Please elaborate.

**A sentence has been added highlighting the fact that increased blade length and O&M costs will both be more pronounced offshore. [Page 3 lines 6-7]**

3. P. 3, Figure 1. As also suggested by Reviewer 1, the naming of radar stations is confusing. The naming could for simplicity be the state names (ID, TX, IL, WI, MI and NY) as you selected only one radar station in each state. For clarity, the full station names could be given once. You have selected a wind farm within a radius of roughly 60 km from each radar station. Would it be possible to add circles of this dimension at each station in the map?

We have added state abbreviations to the 4-letter RADAR station codes throughout the figures and text of the paper. We have elected to keep the 4 digit station codes as well, since these uniquely identify the RADAR stations. Since we have been provided with wind farm data under an NDA and the provider states that their wind farms must not be identified, we have not included the suggested circles around each RADAR station.

4. P. 5, line 1. Hourly precipitation rate (N1P). Did you average to 1 hour or is the data in the database 1-hour data?

Precipitation rates are only 'hourly' in that they have units of mmhr-1. The wording has been changed to avoid this confusion. Thank you. [Page 5 Line 22]

 P. 5, line 8. Is the NCR maybe Normalized Composite Reflectivity? Nearly all of the 3-character codes for general NEXRAD RADAR products begin with an N. I believe it stands for NEXRAD. The list of codes can be found at https://www1.ncdc.noaa.gov/pub/data/radar/RadarProductsDetailedTable.pdf. This gives

NCR as "Composite Reflectivity (16 Levels)"

6. P. 5, line 20. Is the spatial mean based on 5-minute and 6-minute values, then averaged to hourly?

Precipitation rates are calculated every 5-6 minutes. We have updated the text to make this clear. [Page 6 Lines 9-10]

7. P. 6, line 4, m-4. Do not use italic.

- Corrected. Thank you [Page 6 line 14]
- 8. P. 6, line 14, would smaller be more correct than small? Yes. Smaller is correct. This change has been made [Page 7 Line 5]
- 9. P.6, line 16, : : : the slope of the with hydrometeor diameter.. I wonder if "with" is a good formulation here

**This has been edited ('with' removed)**

10. P. 6, line 16: : : is considerably shallower than: : :. Would it be more clear to say nearly constant?

**The fact that the slope is non-zero is important, and the hail number density does decrease by about 2 orders of magnitude over the radius range shown. We are more comfortable with 'shallower'.**

11. Figures 3 to 7. The graphics are good but can be improved. Please use capital letter at the legends at all axis.

**Axis labels and legend entries have been capitalized throughout. Thank you.**

- Fig. 4, upper panel. You show values at 0 mm/hour. Is this indicating all events with no
  precipitation (or from >0 but smaller than 5 mm/hour). Please clarify. Add simplified station
  names.
- Thank you for pointing this out. The lowest tick label has been changed to < 2.5 mm/hr.
- 13. Fig. 5 and Fig. 7. Add simplified station names.

**State abbreviations have been added for each RADAR station.**

- 14. Fig. 6, Add simplified stations names near each of the colored markers, if possible. State abbreviations have been added for each RADAR station here, as well
- 15. Page 9. The discussion of the mean wind speed for all cases at six stations from three information sources and in addition wind speed for rainy cases for six stations from one source. It is a bit confusing. A table summarizing the relevant numbers could be easier for readers to follow the discussion.

Table 2 (Page 12) has been added summarizing the mean wind speed at each site, as well as mean wind speeds conditionally sampled precipitation by the presence or absence of precipitation

16. P.6 line 7 and P. 16, line 2 you write 'conservative'. I am in doubt what you mean in this context. Please clarify.

These estimates are conservative in the engineering sense. They will tend to overestimate damage to blades.

Wording on page 6 has been added to make it clear that a conservative estimate of hail size and probability is an overestimate [Page 6 Line 10-12]

On [Page 17 line 38] (formerly page 16 Line 2), the meaning of the statement that the estimate is conservative is explained parenthetically near the end of the sentence.

**RADAR-Derived Precipitation Climatology for Wind Turbine Blade** Leading Edge Erosion**

**Frederick Letson1 (ORCID: 0000-0001-9275-0359), Rebecca J. Barthelmie2 (ORCID: 0000-0003-0403-6046), Sara C. 5 Pryor1 (ORCID: 0000-0003-4847-3440)**

1Department of Earth and Atmospheric Sciences, Cornell University, Ithaca, New York 2Sibley School of Mechanical and Aerospace Engineering, Cornell University, Ithaca, New York

Correspondence to: F. Letson (fl368@cornell.edu) and S.C. Pryor (sp2279@cornell.edu)

Abstract: Wind turbine blade leading edge erosion (LEE) is a potentially significant source of revenue loss for windfarm
operators. Thus, it is important to advance understanding of the underlying causes, to generate geospatial estimates of erosion potential to provide guidance in pre-deployment planning and ultimately to advance methods to mitigate this effect and extend blade lifetimes. This study focusses on the second issue and presents a novel approach to characterizing the erosion potential across the contiguous USA based solely on publicly available data products from the National Weather Service dual-polarization RADAR. The approach is described in detail and illustrated using six locations distributed across parts of the USA that have

15 substantial wind turbine deployments. Results from these locations demonstrate the high spatial variability in precipitationinduced erosion potential, illustrate the importance of low probability high impact events to cumulative annual total kinetic energy transfer and emphasize the importance of hail as a damage vector.

**1 Introduction and objectives**

- In 2017 wind turbines (WT) provided 6% of total electricity generation in the United States of America (USA) (U.S. 20 Energy Information Administration, 2018) and there are over 50,000 WT operating in the USA today (Pryor et al., 2019). WT are subject to harsh operating conditions during their 20-25 year lifetimes including; extreme winds, impacts from heavy rain, hailstones and snow, and intense ultraviolet light exposure that can lead to material damage (Keegan et al., 2013). Accordingly, operation and maintenance (O&M) costs comprise 20-25% of the total levelized cost per kWh of electricity produced over the WT lifetime (Mishnaevsky Jr, 2019; Moné et al., 2017). WT blades exhibit the highest failure rate (FR ~ 0.2) of any WT
- 25 component (Zhu and Li, 2018). The most expensive repair and longest repair times are associated with blades (Shohag et al., 2017). Estimates suggest average cost of blade repair of an onshore turbine is approximately \$30,000, with replacement costs of ~ \$200,000 (Mishnaevsky Jr, 2019). Repair and replacement costs will tend to be higher offshore where general O&M costs are higher (~30% of total cost) and blade failures contribute also significantly to turbine downtime (Carroll et al., 2016). Hail has long been recognized as an important source of weather-related economic losses in the Contiguous United States
- 30 (CONUS) (Changnon, 1999; Cintineo et al., 2012). Economic losses from hail were estimated to be \$1.2 billion in 1999 (Changnon, 1999), and property damage from severe hail has been shown to be increasing with time (Changnon, 2009), with more recent annual losses estimated at \$10 billion, accounting for almost 70% of severe-weather-related insurance claims (Loomis, 2018). An analysis conducted in 2009 indicated an average of 159 days each year are associated with crop-damaging hail leading to average crop loss of \$580 million, and hail damage to property was valued at \$850 million (Changnon et al.,
- 35 2009). Hail, and hail damage, are highly episodic. For example, insurance losses in the Dallas–Fort Worth (DFW) metroplex on

1

a single hail day in May 2011 were estimated to exceed \$876 million (Brown et al. 2015). While the paucity and subjectivity of observed hail data sets make a global comparison difficult, severe hail is almost certainly more common in the central US than in other areas of the world with substantial wind energy development (Prein and Holland, 2018). Further, the relationship linking, damage to the frequency and severity of hail varies substantially with the target, WT present an interesting challenge in this context because they are large structures and the blades are rotating, composite materials.

A key cause of the need for WT blade repairs is excess damage (i.e. material loss) on the leading edge (leading edge erosion, LEE). LEE roughens WT blades, reducing lift and electrical power production (Sareen et al., 2014; Gaudern, 2014). LEE causes an average of 1-5% reduction in annual energy production (AEP) (Froese, 2018) and up to a 9% reduction when delamination occurs (Schramm et al., 2017). Thus, excess LEE may be costing the industry tens of millions of dollars per year

5

- 10 via lost revenue and/or increased maintenance costs, and poses a threat to achieving continuing wind energy cost reductions (Sareen et al., 2014). In response to this issue a major industrial research consortium from Europe (including DNV GL, Vestas and Siemens Gamesa Renewable Energy) has recently (Nov 2018) announced a new partnership (COBRA) focused on analysis of mitigation measures for LEE including development of next-generation leading-edge protection systems (Durakovic, 2019).
- WT blades use composites (e.g. epoxy or polyester, with reinforcing glass or carbon fibers) (Mishnaevsky et al., 2017)
   coated to protect the blade structure by distributing and absorbing the energy from impacts (Brøndsted et al., 2005). Thus, the leading edge actually comprises several layers of the main structural composite material (and thickening materials) plus coatings (Mishnaevsky et al., 2017). Impact fatigue caused by collision with rain droplets and hail stones is a primary cause of WT blade LEE (Bech et al., 2018; Bartolomé and Teuwen, 2019; Zhang et al., 2015). Although rain droplets fall at only modest velocities (typically ≤ 10 ms-1, see details below), the tip of WT blades rotate quickly (50-110 ms-1), thus the net closing velocity and
- 20 kinetic energy transfer are large. Each precipitation impact on the blade leading edge results in transient stresses that are proportional to impact velocity (Preece, 1979; Slot et al., 2015). The stress induced by individual high net collision impacts with hydrometeors may, in principle, exceed the strength of the material. Estimates of the failure energy threshold of a composite structure vary widely (e.g. values of 72 140 J are given in (Appleby-Thomas et al., 2011)) and may exceed 300 J for leading-edge thicknesses and hailstone diameters > 20 mm (Kim and Kedward, 2000). However, conceptually the erosion of
- 25 homogeneous materials is most frequently considered using a three stage model. Initially there is an incubation period during which impacts occur but no visible damage is observed although microstructural changes in the materials generate nucleation sites for material removal which commences when a threshold is reached (i.e. when some level of accumulated impacts is reached). Once the time to damage has been exceeded additional damage occurs as stress waves propagate from the impact sites into the composite and cause existing pits and cracks to grow and there is a steady increase of material loss occurs with each
- 30 additional impact (Cortés et al., 2017; Eisenberg et al., 2018; Traphan et al., 2018). The number of impacts required to reach the threshold for surface fatigue failure is a function of the droplet diameter and phase, the closing velocity, the strength of the material and the pressure of the impact. Hence, the materials response to hail (solid hydrometeors) may differ from that to collisions with liquid (rain) droplets. For example, the maximum von-Mises stress created in the WT blade leading edge from a 10 mm diameter hailstone greatly exceed that from a rain droplet of equivalent size and closing velocity due to differences in mass and hardness (Keegan et al., 2013).

WT LEE is a developing area of research and uncertainty remains regarding frequency and severity of the issue. Rates of LEE appear to be highly spatially variable due to variations in WT operating conditions and the precipitation climate. Industrial experience has demonstrated exposure to particularly harsh operating conditions can erode coatings causing partial delamination after as little as 2-3 years (Rempel, 2012; Keegan et al., 2013). Elastomeric coatings can be applied for additional erosion

40 resistance (Dalili et al., 2009; Valaker et al., 2015; Herring et al., 2019). However, the life of such coatings cannot be predicted

**Deleted:** Estimates of total annual hail damage to property and to crops in the Contiguous United States (CONUS) exceed \$1 billion, and threlationship**

**Deleted: of**

**Deleted: with the type of target**

**Deleted: Being**

**,**

accurately (and is a function of UV exposure, (Shokrieh and Bayat, 2007)), they have a negative impact on blade aerodynamics (Giguère and Selig, 1999) and their cost-effectiveness is uncertain (Dashtkar et al., 2019).

The total installed capacity (IC), rated capacity (and physical dimensions) of  $_{\psi}WT_{\psi}$  being installed exhibited marked growth in the USA over the last 20 years (Wiser and Bolinger, 2018; Wiser et al., 2016). Average WT blade length increased from < 4 m in 1985 to 32 m in 2005 and now exceeds 55 m (Wiser and Bolinger, 2018). Since the tip speed increases with blade length, this

- tendency towards taller WT with longer blades exacerbates LEE potential. The increased blade length and larger maintenance costs associated with offshore wind turbines tend to make offshore wind farms especially vulnerable to LEE. Based on previous research the *a priori* expectation of this research is that excess LEE is most likely on WT deployed in environments with high rain intensities and hail frequencies such are experienced in the Great Plains (the states of Texas (TX), Oklahoma (OK), Kansas
- 10 (KS), Nebraska (NE), North and South Dakota (ND, SD), Wyoming (WY) and Montana (MN); Fig. 1). LEE is likely to present a growing issue within the US wind industry as more and larger wind turbines with higher tip-speed ratios are deployed (Amirzadeh et al., 2017a). The current average age of WT in the US is 9 years (AWEA, 2019) and LEE will be of greater concern as a larger number of WT move out of the typical 1 to 5 year warranty period (Bolinger and Wiser, 2012; Brown, 2010).
- Addressing the challenges posed by blade LEE and developing mitigation options requires multi-scale and multidisciplinary research. Given the importance of precipitation phase, size and intensity during WT operation to the potential for blade LEE here we focus on developing a consistent and generalizable framework that can be applied to derive estimate of erosion-relevant atmospheric properties. We present an objective, spatially consistent, robust and repeatable framework that can be applied across CONUS and crucially uses only non-commercial (i.e. publicly available) data. The specific objectives of the research reported herein are:
- 20 1) To develop the workflow necessary to develop a proto-type RADAR-based erosion atlas.
  - To provide a first estimate of the spatial variability of erosion potential across CONUS in regions where wind turbines are currently deployed (see Fig. 1).
  - To conduct an initial uncertainty propagation exercise to illustrate how uncertainties in the input data propagate through the analysis workflow to influence erosion potential estimates.
- 25 4) To describe the degree to which blade LEE is episodic and therefore amendable to the mitigation strategy proposed earlier in research from Denmark of WT curtailment during 'highly erosive' periods. The efficacy of this strategy is a function of (i) the wind speed regime and joint probability distributions of erosive events (heavy rain or hail) and power-producing wind speeds, (ii) price of electricity supplied to the grid and (iii) O&M costs. A cost-benefit analysis based on conditions in Denmark suggested the loss of revenue from the curtailment of power production was small compared to the economic
- 30 benefits from enhance blade lifetimes (Bech et al., 2018).

5

| -[ | Deleted: wind turbines (             |
|----|--------------------------------------|
| (  | Deleted: )                           |
|    |                                      |
|    |                                      |
|    |                                      |
|    |                                      |
|    |                                      |
| -[ | Deleted: Central                     |
|    |                                      |
|    |                                      |
|    |                                      |
|    |                                      |
| _  |                                      |
| -( | Formatted: Indent: First line: 0.25" |
|    |                                      |
|    |                                      |
|    |                                      |
| _  |                                      |
| 1  | Deleted: the continental             |

| -{ | Deleted: While t                                                                                  |
|----|---------------------------------------------------------------------------------------------------|
| (  | Deleted: also                                                                                     |
| (  | Deleted: sensitive to                                                                             |
| (  | Deleted: and cost of energy                                                                       |
| ſ  | Deleted: the lost energy due to speed curtailment is small compared with the losses of LEE |

Figure 1 Locations of wind turbines as deployed at the end of 2017 according the UGSGS database (available from; https://eerscmap.usgs.gov/uswtdb/) (grey dots), the NWS RADAR stations from which data are presented (see details in Table 1), and areas of frequent hail occurrence. Areas with more than nine hail days per year are outlined by the red contour, and those with more than six are outlined by the blue contours (Cintineo et al., 2012).

Table 1: The station code and locations of the six NWS dual polarization RADARs from which data are presented (listed from west to east).

| Station code | Latitude (N) | Longitude (E) | State |
|--------------|--------------|---------------|-------|
| KSFX         | 43.106       | -112.686      | ID    |
| KMAF         | 31.943       | -102.189      | TX    |
| KDVN         | 41.612       | -90.581       | IA    |
| KMKX         | 42.968       | -88.551       | WI    |
| KGRR         | 42.894       | -85.545       | MI    |
| KBUF         | 42.949       | -78.737       | NY    |

**Deleted: IL**

**2 Data and Methods**

5

I

10 A first estimate of precipitation-derived erosion potential at sites across the USA as developed in the current work is based on a characterization of the kinetic energy exchange from rain and hail impacts on the blade leading edge. The procedure used in making these estimates is divided into two steps: Calculation of meteorological parameters (wind speed, rain and hail) at six wind farms, each located within the observation area, of a RADAR station, and then calculation of blade impact frequencies and energy transfer based on those meteorological parameters. Exact wind farm locations and details are excluded from this paper junder a non-disclosure agreement (NDA).

| Deleted: a                                          |
|-----------------------------------------------------|
| Deleted: nominal                                    |
| Deleted: s                                          |
| Deleted: six                                        |
| Deleted: RADARs                                     |
| Deleted: to protect developer data acquired under a |

The research reported herein leverages resources generated from the upgraded National Weather Service (NWS) network of WSR-88D RADAR to dual polarization (completed in 2013, (Seo et al., 2015; Crum et al., 1998)) along with the NOAA Weather and Climate Toolkit (WCT) (see details of the data products and data volumes provided in Appendix A). These data represent a unique opportunity to characterize precipitation properties such as hail that are very challenging to detect and to

- 5 accurately characterize using in situ methods or human observers (see discussion in (Allen and Tippett, 2015) and details of RADAR operation (Kumjian, 2018)). NWS RADAR operate at elevation angles between 0.5° and 19.5° and an azimuthal resolution of 1°. Doppler and dual-polarization data are publicly available at a resolution of 0.25 km up to a range of 300 km from each RADAR site (NOAA, 1991; Istok et al., 2009) (see description of the data provision in (Kelleher et al., 2007) and an example of the NWS products given in Fig. 2). The temporal resolution of the data is typically ~ 5 minutes, but varies slightly
- 10 with scanning mode: 1) Clear Air Mode uses longer, 10-minute scans to collect sufficient return data during times of no precipitation when signal return strength is relatively low. 2) Precipitation mode is used when there is any precipitation detected in the scan area and uses a 6-minute scan cycle. 3) Storm mode is used when severe or rapidly-evolving storms are present, and uses a 5-minute sampling interval, made possible by reducing the number of elevation angles used (NOAA, 2016a). Storm detection and tracking using RADAR is a complex and evolving science but in brief the NWS system uses an automated function
- 15 which employs reflectivity from the current scan and storm cell location and vertically integrated liquid water (VIL) from the previous scan (Johnson et al., 1998).

To illustrate the proposed analysis framework we use data from six NWS dual-polarization Doppler RADAR stations (see Fig. 1 and Table 1) collected over the period 2014-2018. These locations were chosen to represent gradients in hail probability and precipitation amount in regions with relatively high wind turbine installed densities (Fig. 1). We employ the framework in order to generate erosion climates for six wind farms operating in the scanned volume of the RADARs and located 35-75 km

- from the RADAR locations. The following RADAR data products are used (see also Appendix A):
  - Precipitation rate (N1P) is the precipitation rate in each RADAR cell in each ~ 5 minute period (expressed in units of mmhr-1) as estimated from reflectivity.

25

20

- Hybrid Hydrometer Classification (HHC): Based on reflectivity, temperature, and dual polarization variables, HHC is an estimate of the most likely targets within the RADAR volume. While this is derived product, classification algorithms and accuracy have improved with the widespread adoption of dual polarization RADAR and application of areal (rather than point-wise) techniques (NOAA, 2016b; Chandrasekar et al., 2013). The hydrometeor types encoded in the NWS data product are; dry snow, wet snow, crystals, big drop, rain (light and moderate), heavy rain, graupel, and rain with hail.
   Hail reports (NHI): Maximum hail size (an estimate of the 75th percentile hail stone diameter (*D75*)) and probability of hail

- 30 are used to identify the occurrence and severity of hail events (see discussion in (Witt et al., 1998)).
  - Composite Reflectivity (NCR): Maximum reflectivity at any elevation angle measured in each RADAR cell. This is used here to characterize the spatial extent of hail events (i.e. reflectivity > 50 dBZ (Witt et al., 1998)).
  - Radial wind speeds from the 0.5° elevation angle as computed from the Doppler shift (N0V) (Alpert and Kumar, 2007).

| 1 | Deleted: nominal  |
|---|-------------------|
|   |                   |
| ( | Deleted: Hourly p |
| ſ | Deleted:          |

---

## Author Response (AR1)

REPSONSE TO REVIEWERS

RADAR-Derived Precipitation Climatology for Wind Turbine Blade Leading Edge Erosion

F. Letson, R.J. Barthelmie, S.C. Pryor

MS No.: wes-2019-43

MS Type: Research article

Special Issue: Wind Energy Science Conference 2019

**Julio J. Melero (Editor) melero@unizar.es

Dear authors, please post your response to the reviewers comments.

I kindly ask you to address the point recommendations given by both reviewers in order to let me evaluate in detail the modifications that you incorporate in the final paper.

**The authors would like to thank the two anonymous reviewers for their helpful comments. Our responses are enumerated below in bold, aligned to the right side of the page. In cases where the line number of an edit has changed from the previous version of the manuscript, the new line number is included [in square brackets]. A tracked-changes version of the manuscript is included, below.**

**Anonymous Referee #1**
**General Comments**
This paper computes the energy of impact to wind turbines by rain and hail, a topic that is seldom discussed, yet very important to operation and maintenance of wind farms. It points out the potential for these hydrometeors to erode the leading edge of wind turbines. They leverage public data for the US regarding dual-pol radar data and estimate the energy for typical turbine parameters. This paper should be of interest to readers of WES. The work is well-justified, interesting, and well presented. The authors justify that the problem is important and review the subject appropriately. The calculations are appropriate, but quite conservative, as the tip speed is used in computing the energy of impact. In fact, the tip itself would be seldom hit. It may be useful to provide a range of impacts as a function of distance from the hub.

**Specific Comments (page and line numbers refers to tracked changes version of the manuscript)**
1. The literature review is appropriate, but one wonders how much similar research has been done by the insurance industry regarding damage to cars and roofs.

   **This is an excellent point. New text (and references) has been added to the introduction that documents economic losses for the USA [Page 1 Lines 27 to Page 2 Line 5]**

2. This paper focuses on the U.S. It would be helpful to discuss how it is expected to apply in other parts of the world.

   **The extent of RADAR coverage and the frequency of hail are both reasons why a study like this one makes particular sense in the US. We have added a sentence (and a reference) about the comparatively high frequency of hail events in the central US [Page 2 Lines 1-3] and information about broader applicability (e.g. to the European RADAR network to the conclusions [Page 17-18]**

3. P. 3, line 11 – reference to the work of Bech et al. for Denmark on curtailment is interesting. Did they do a cost-benefit analysis? Readers may be interested.

   **Yes, they did. Bech et al. found their curtailment strategy to be cost effective. This has been added to the introduction [page 3 line 26-30]**

4. P. 5, line 5 – It would be interesting to comment on the accuracy of the current classification methods of the dual pol radars – it used to be rather poor, but may have improved with time.

   **Good point. A sentence on improvements in classification has been added. [Page 5 line 25-27]**

5. P. 6, line 7 – Please clarify what you mean by the "largest values". Do you mean the 75% from the prior page? Later on line 15, you refer to a fitting parameter for D_75, which suggests that.

**Yes. Thank you. This has been made explicit. [Page 6 Line 17-19]**

6. P. 6, line 29 – "the mean RPM begins to decrease at wind speeds below the cut-out velocity …" Is this true? Please provide references. The power remains constant at rated capacity until nearly at cut-out, so how does the RPM decrease? Or do you mean as it approaches cut-out speed? Please clarify

**This is based on our analysis of SCADA data. There are a significant number of below-rated-RPM events for wind speeds approaching cutout. This passage has been re-worded to clarify. [Page 7 Line 21-22]**

7. P. 7, line 6 – The rest of the analysis uses the speed of the blade tip for calculations. What if you used the mid-point of the blade instead as a more representative speed? You do mention that this is "conservative". It would be informative to compare the speed at the mid-point to help show the variability across the blade. Or even to plot impact as a function of distance from the hub.

**Leading edge erosion is primarily of concern at or near the blade tip. An explanation (including the fact that local blade speeds vary linearly with distance from the hub) and reference have been added. [Page 7 Lines 36-38]**

8. P. 15 – lines 19-p. 16, line 10 - -This is a nice list of limitations. Thanks for providing.

**Thank you.**

**Technical corrections**

9. P. 1, line 26 – please write out "approximately" (not approx..)

**Done. Thank you.**

10. Throughout – please add space between references in parentheses

**Done**

11. Throughout – it is difficult to follow and remember the 4-letter designations for the radar sites, even for an American reviewer. It will likely be even more difficult for international readers. Perhaps using the states where they are located in your references to the sites would enhance readability??

**Good idea. We have added state abbreviations to the RADAR station codes throughout the text and figures.**

12. P. 2, line 30 – WT was already defined

**Thank you 'WT' is now used here without the redundant definition [Page 3 Line 3]**

13. P. 2, line 35 – please specify U.S. Central Plains – this in an international journal.

**Thank you, this has been added [Page 3 lines 9-10]**

14. P. 3, line 6 – please define CONUS first time it is used.

**Done [Page 1 Line 29-30].**

15. P. 4, line 9 – "nominal wind farm located within the observation areas of six RADARS" is confusing. Are you referring to a wind farm for each of the six RADARS? What do you mean by "nominal" wind farm? Is this "notional"? Have you identified a specific wind farm or are you referring to one within the reach of the radar beam? As written, it implies that you have identified a wind farm that is reached by the beams of all 6 radars, which is surely not what you meant to say.

**Thank you, the wording has been changed to make it clear that there 6 actual wind farms, each covered by a RADAR station [Page 4 lines 12-15]**

16. P. 5, line 17 – I don't think you mean to refer to Fig. 3 here.

**This reference to Fig 3 has been removed [Page 6 Line 7]**

17. P. 6, line 11 – "hailstones" should be plural

**This has been corrected [Page 7 line 1]**

18. P. 6, line 13 – "sampled IN Alberta"

**Yes, thank you. Done. [Page 7 Line 4]**

19. P. 6, line 27 – "wind speed AS shown in …"

**Done. [Page 7 Line 18]**

20. P. 11, line 10 – hail storms are quite frequent in Boulder, Co which is just west of 105°

**'Very infrequent' has been softened to 'much less frequent', which is still consistent with the literature cited (Cintineo et al and Allen and Tippett) [Page 12, Line 20]**

21. P. 12, line 7 – occurring in fewer that …" (not few). The point you make here is certainly true for damage to roofs and cars.

**Thank you. This has been corrected [Page 14, line 5]**

22. P. 15, line 5 – " it is flexible to use with different …" (not "of")

**Corrected. [Page 17, line 2]**

23. P. 15, line 12 – please indent paragraphs.

**Done. Thank you. [Page 17, Line 11]**

24. P. 16, lines 5-7 – The word "herein" is used 3 times in 3 lines.

**The wording has been changed to avoid this repetition. Thank you. [Page 17, Lines 26 to 29]**

**Anonymous Referee #2**

The paper by Letson et al. presents an interesting study on an emerging research field related to wind energy. It is of high value for readers of WES. The methodology is presented in a clear way. The results are discussed at a high level with many papers cited within the cross-cutting field. The results are novel. The conclusion summarizes in outstanding way the many learnings and perspectives for further research. The paper is recommended for publication.

**Minor edits (page and line numbers refers to tracked changes version of the manuscript)**

1. P.1, line 26. Would the repair cost be different between on- and offshore wind farms? Please clarify.

**Text (and a reference) has been added about the higher cost of maintenance for offshore turbines [Page 1 lines 27-28]**

2. P. 2, line 38. Another challenge than age alone could be US East Coast wind farms. Please elaborate.

**A sentence has been added highlighting the fact that increased blade length and O&M costs will both be more pronounced offshore. [Page 3 lines 6-7]**

3. P. 3, Figure 1. As also suggested by Reviewer 1, the naming of radar stations is confusing. The naming could for simplicity be the state names (ID, TX, IL, WI, MI and NY) as you selected only one radar station in each state. For clarity, the full station names could be given once. You have selected a wind farm within a radius of roughly 60 km from each radar station. Would it be possible to add circles of this dimension at each station in the map?

**We have added state abbreviations to the 4-letter RADAR station codes throughout the figures and text of the paper. We have elected to keep the 4 digit station codes as well, since these uniquely identify the RADAR stations. Since we have been provided with wind farm data under an NDA and the provider states that their wind farms must not be identified, we have not included the suggested circles around each RADAR station.**

4. P. 5, line 1. Hourly precipitation rate (N1P). Did you average to 1 hour or is the data in the database 1-hour data?

**Precipitation rates are only 'hourly' in that they have units of mmhr$^{-1}$. The wording has been changed to avoid this confusion. Thank you. [Page 5 Line 22]**

5. P. 5, line 8. Is the NCR maybe Normalized Composite Reflectivity?

**Nearly all of the 3-character codes for general NEXRAD RADAR products begin with an N. I believe it stands for NEXRAD. The list of codes can be found at https://www1.ncdc.noaa.gov/pub/data/radar/RadarProductsDetailedTable.pdf. This gives NCR as "Composite Reflectivity (16 Levels)"**

6. P. 5, line 20. Is the spatial mean based on 5-minute and 6-minute values, then averaged to hourly?

**Precipitation rates are calculated every 5-6 minutes. We have updated the text to make this clear. [Page 6 Lines 9-10]**

7. P. 6, line 4, m-4. Do not use italic.

**Corrected. Thank you [Page 6 line 14]**

8. P. 6, line 14, would smaller be more correct than small?

**Yes. Smaller is correct. This change has been made [Page 7 Line 5]**

9. P.6, line 16, : : :the slope of the with hydrometeor diameter.. I wonder if "with" is a good formulation here

**This has been edited ('with' removed)**

10. P. 6, line 16: : :is considerably shallower than: : :. Would it be more clear to say nearly constant?

**The fact that the slope is non-zero is important, and the hail number density does decrease by about 2 orders of magnitude over the radius range shown. We are more comfortable with 'shallower'.**

11. Figures 3 to 7. The graphics are good but can be improved. Please use capital letter at the legends at all axis.

**Axis labels and legend entries have been capitalized throughout. Thank you.**

12. Fig. 4, upper panel. You show values at 0 mm/hour. Is this indicating all events with no precipitation (or from >0 but smaller than 5 mm/hour). Please clarify. Add simplified station names.

**Thank you for pointing this out. The lowest tick label has been changed to < 2.5 mm/hr.**

13. Fig. 5 and Fig. 7. Add simplified station names.

**State abbreviations have been added for each RADAR station.**

14. Fig. 6, Add simplified stations names near each of the colored markers, if possible.

**State abbreviations have been added for each RADAR station here, as well**

15. Page 9. The discussion of the mean wind speed for all cases at six stations from three information sources and in addition wind speed for rainy cases for six stations from one source. It is a bit confusing. A table summarizing the relevant numbers could be easier for readers to follow the discussion.

**Table 2 (Page 12) has been added summarizing the mean wind speed at each site, as well as mean wind speeds conditionally sampled precipitation by the presence or absence of precipitation**

16. P.6 line 7 and P. 16, line 2 you write 'conservative'. I am in doubt what you mean in this context. Please clarify.

**These estimates are conservative in the engineering sense. They will tend to overestimate damage to blades.**

**Wording on page 6 has been added to make it clear that a conservative estimate of hail size and probability is an overestimate [Page 6 Line 10-12]**

**On [Page 17 line 38] (formerly page 16 Line 2), the meaning of the statement that the estimate is conservative is explained parenthetically near the end of the sentence.**

**RADAR-Derived Precipitation Climatology for Wind Turbine Blade Leading Edge Erosion**

**Frederick Letson[1] (ORCID: 0000-0001-9275-0359)**, **Rebecca J. Barthelmie[2] (ORCID: 0000-0003-0403-6046)**, **Sara C. Pryor[1] (ORCID: 0000-0003-4847-3440)**

[1]Department of Earth and Atmospheric Sciences, Cornell University, Ithaca, New York

[2]Sibley School of Mechanical and Aerospace Engineering, Cornell University, Ithaca, New York

*Correspondence to*: F. Letson (fl368@cornell.edu) and S.C. Pryor (sp2279@cornell.edu)

**Abstract:** Wind turbine blade leading edge erosion (LEE) is a potentially significant source of revenue loss for windfarm operators. Thus, it is important to advance understanding of the underlying causes, to generate geospatial estimates of erosion potential to provide guidance in pre-deployment planning and ultimately to advance methods to mitigate this effect and extend blade lifetimes. This study focusses on the second issue and presents a novel approach to characterizing the erosion potential across the contiguous USA based solely on publicly available data products from the National Weather Service dual-polarization RADAR. The approach is described in detail and illustrated using six locations distributed across parts of the USA that have substantial wind turbine deployments. Results from these locations demonstrate the high spatial variability in precipitation-induced erosion potential, illustrate the importance of low probability high impact events to cumulative annual total kinetic energy transfer and emphasize the importance of hail as a damage vector.

**1    Introduction and objectives**

In 2017 wind turbines (WT) provided 6% of total electricity generation in the United States of America (USA) (U.S. Energy Information Administration, 2018) and there are over 50,000 WT operating in the USA today (Pryor et al., 2019). WT are subject to harsh operating conditions during their 20-25 year lifetimes including; extreme winds, impacts from heavy rain, hailstones and snow, and intense ultraviolet light exposure that can lead to material damage (Keegan et al., 2013). Accordingly, operation and maintenance (O&M) costs comprise 20-25% of the total levelized cost per kWh of electricity produced over the WT lifetime (Mishnaevsky Jr, 2019; Moné et al., 2017). WT blades exhibit the highest failure rate (FR ~ 0.2) of any WT component (Zhu and Li, 2018). The most expensive repair and longest repair times are associated with blades (Shohag et al., 2017). Estimates suggest average cost of blade repair of an onshore turbine is approximately $30,000, with replacement costs of ~ $200,000 (Mishnaevsky Jr, 2019). Repair and replacement costs will tend to be higher offshore where general O&M costs are higher (~30% of total cost) and blade failures contribute also significantly to turbine downtime (Carroll et al., 2016).

Hail has long been recognized as an important source of weather-related economic losses in the Contiguous United States (CONUS) (Changnon, 1999; Cintineo et al., 2012). Economic losses from hail were estimated to be $1.2 billion in 1999 (Changnon, 1999), and property damage from severe hail has been shown to be increasing with time (Changnon, 2009), with more recent annual losses estimated at $10 billion, accounting for almost 70% of severe-weather-related insurance claims (Loomis, 2018). An analysis conducted in 2009 indicated an average of 159 days each year are associated with crop-damaging hail leading to average crop loss of $580 million, and hail damage to property was valued at $850 million (Changnon et al., 2009). Hail, and hail damage, are highly episodic. For example, insurance losses in the Dallas–Fort Worth (DFW) metroplex on

a single hail day in May 2011 were estimated to exceed \$876 million (Brown et al. 2015). While the paucity and subjectivity of observed hail data sets make a global comparison difficult, severe hail is almost certainly more common in the central US than in other areas of the world with substantial wind energy development (Prein and Holland, 2018). Further, the relationship linking damage to the frequency and severity of hail varies substantially with the target. WT present an interesting challenge in this

5 context because they are large structures and the blades are rotating, composite materials.

A key cause of the need for WT blade repairs is excess damage (i.e. material loss) on the leading edge (leading edge erosion, LEE). LEE roughens WT blades, reducing lift and electrical power production (Sareen et al., 2014; Gaudern, 2014). LEE causes an average of 1-5% reduction in annual energy production (AEP) (Froese, 2018) and up to a 9% reduction when delamination occurs (Schramm et al., 2017). Thus, excess LEE may be costing the industry tens of millions of dollars per year

10 via lost revenue and/or increased maintenance costs, and poses a threat to achieving continuing wind energy cost reductions (Sareen et al., 2014). In response to this issue a major industrial research consortium from Europe (including DNV GL, Vestas and Siemens Gamesa Renewable Energy) has recently (Nov 2018) announced a new partnership (COBRA) focused on analysis of mitigation measures for LEE including development of next-generation leading-edge protection systems (Durakovic, 2019).

WT blades use composites (e.g. epoxy or polyester, with reinforcing glass or carbon fibers) (Mishnaevsky et al., 2017)

15 coated to protect the blade structure by distributing and absorbing the energy from impacts (Brøndsted et al., 2005). Thus, the leading edge actually comprises several layers of the main structural composite material (and thickening materials) plus coatings (Mishnaevsky et al., 2017). Impact fatigue caused by collision with rain droplets and hail stones is a primary cause of WT blade LEE (Bech et al., 2018; Bartolomé and Teuwen, 2019; Zhang et al., 2015). Although rain droplets fall at only modest velocities (typically $\leq$ 10 ms$^{-1}$, see details below), the tip of WT blades rotate quickly (50-110 ms$^{-1}$), thus the net closing velocity and

20 kinetic energy transfer are large. Each precipitation impact on the blade leading edge results in transient stresses that are proportional to impact velocity (Preece, 1979; Slot et al., 2015). The stress induced by individual high net collision impacts with hydrometeors may, in principle, exceed the strength of the material. Estimates of the failure energy threshold of a composite structure vary widely (e.g. values of 72 – 140 J are given in (Appleby-Thomas et al., 2011)) and may exceed 300 J for leading-edge thicknesses and hailstone diameters > 20 mm (Kim and Kedward, 2000). However, conceptually the erosion of

25 homogeneous materials is most frequently considered using a three stage model. Initially there is an incubation period during which impacts occur but no visible damage is observed although microstructural changes in the materials generate nucleation sites for material removal which commences when a threshold is reached (i.e. when some level of accumulated impacts is reached). Once the time to damage has been exceeded additional damage occurs as stress waves propagate from the impact sites into the composite and cause existing pits and cracks to grow and there is a steady increase of material loss occurs with each

30 additional impact (Cortés et al., 2017; Eisenberg et al., 2018; Traphan et al., 2018). The number of impacts required to reach the threshold for surface fatigue failure is a function of the droplet diameter and phase, the closing velocity, the strength of the material and the pressure of the impact. Hence, the materials response to hail (solid hydrometeors) may differ from that to collisions with liquid (rain) droplets. For example, the maximum von-Mises stress created in the WT blade leading edge from a 10 mm diameter hailstone greatly exceed that from a rain droplet of equivalent size and closing velocity due to differences in

35 mass and hardness (Keegan et al., 2013).

WT LEE is a developing area of research and uncertainty remains regarding frequency and severity of the issue. Rates of LEE appear to be highly spatially variable due to variations in WT operating conditions and the precipitation climate. Industrial experience has demonstrated exposure to particularly harsh operating conditions can erode coatings causing partial delamination after as little as 2-3 years (Rempel, 2012; Keegan et al., 2013). Elastomeric coatings can be applied for additional erosion

40 resistance (Dalili et al., 2009; Valaker et al., 2015; Herring et al., 2019). However, the life of such coatings cannot be predicted

[revised manuscript text omitted]

- Precipitation rate (N1P) is the precipitation rate in each RADAR cell in each ~ 5 minute period (expressed in units of mmhr⁻¹) as estimated from reflectivity.

- Hybrid Hydrometer Classification (HHC): Based on reflectivity, temperature, and dual polarization variables, HHC is an
25     estimate of the most likely targets within the RADAR volume. While this is derived product, classification algorithms and accuracy have improved with the widespread adoption of dual polarization RADAR and application of areal (rather than point-wise) techniques (NOAA, 2016b; Chandrasekar et al., 2013). The hydrometeor types encoded in the NWS data product are; dry snow, wet snow, crystals, big drop, rain (light and moderate), heavy rain, graupel, and rain with hail.

[revised manuscript text omitted]
. For example, the Network of European Meteorological Services (EUMETNET) operates over 200 RADAR many of which have been upgraded to dual polarization (Saltikoff et al., 2018). The tool proposed here could be used to provide a first assessment of the erosion climate in which a given sited turbine may operate in. It thus provides an important first step towards enabling an assessment of the threat of excessive precipitation-induced LEE in a given deployment environment and the cost-effectiveness of options to reduce the likelihood of

10   premature blade damage.

    The actual likelihood of excess WT LEE and blade damage in any environment is not only a function of the precipitation and wind climate, but also the WT dimensions, materials used in the blade coatings and the coating thickness (Eisenberg et al., 2018; Slot et al., 2015), the presence of existing micro-structural defects (Evans et al., 1980) due to manufacturing defects and damage during transportation (Keegan et al., 2013; Nelson et al., 2017) and other aspects of the operating environment

15   (including thermal fatigue and the occurrence of icing (Slot et al., 2015)).

    The preliminary estimates of erosion potential and the partitioning between liquid precipitation and hail are naturally subject to limitations including, in likely order of importance:

- The relatively short duration of time for which the dual-polarization RADAR products are available. The upgrade of the NWS RADAR network to dual polarization was completed in April 2013, thus only the complete years of 2014-2018,

20   inclusive were available for analysis. Given the large inter-annual variability in precipitation climates this is too short to build a comprehensive climatology (Karl et al., 1995; Prein and Holland, 2018). Any geospatial depiction of the potential precipitation erosion climate will vary according to the precise data period used to compute the climatology and may evolve as a result of climate non-stationarity altering aspects of the precipitation climate (e.g. probability of hail (Brimelow et al., 2017) and rainfall intensity (Easterling et al., 2000)).

25   - The applicability of the RADAR-derived wind speed estimates to derive wind turbine blade rotational speed. There are considerable challenges to line-of-sight wind retrievals from RADAR (Fast et al., 2008). The approach adopted herein assumes a uniform wind flow pattern to derive the wind speeds at the nominal wind turbine hub-height which may not be realized. As described above, while the wind speed climates at five of the six locations considered exhibited relatively good agreement with previous estimates of wind climates, values for the location in Texas are negatively biased. This likely

30   results in a negative bias in kinetic energy transfer for this site.
- Assumptions regarding the size distribution, occurrence and terminal velocities of hail (Dessens et al., 2015; Allen et al., 2017; Heymsfield et al., 2014). The evolution of the NWS RADAR network to dual polarization provides an unprecedented opportunity for spatial estimates of hail presence and size in clouds (Kumjian et al., 2018). However, hail production is a complex and incompletely understood phenomenon (Dennis and Kumjian, 2017; Blair et al., 2017; Pruppacher and Klett,

35   2010). There are substantial event-to-event variations in the size distribution and density of hail stones (Heymsfield et al., 2014), in the presence of solid-phase hydrometeors in clouds (as detected by RADAR) and the occurrence of hail at the ground (Kumjian et al., 2019). Estimates of hail occurrence, size distribution and terminal fall velocity presented herein are likely conservative (i.e. upper bounds on true values), and thus LEE may be over-estimated.
- Assumptions regarding the size distribution of rain droplets. Most observational studies indicate an exponential form

40   (Uijlenhoet, 2001), and the Marshall-Palmer distribution is the most widely applied. However, a range of different forms

have been proposed to describe the size spectrum of rain droplets (*dN/dR*) including gamma (Ulbrich, 1983) and lognormal (Feingold and Levin, 1986), an alternative exponential form (Best, 1950) and more complex non-parametric forms (Morrison et al., 2019). There is also evidence that droplet size distributions may exhibit a functional dependence on near-surface wind speed (Testik and Pei, 2017).

5     •   Assumptions applied in deriving precipitation intensity and other precipitation properties from RADAR. Notable event-to-event variations in the applicability of Z-R relationships have been reported during rain (Uijlenhoet, 2001; Villarini and Krajewski, 2010).

Future work could address and reduce these uncertainties and adapt this approach to examine different wind turbines (by applying a different RPM curve) and/or to assimilate different atmospheric data and/or incorporate more explicit aspects of

10  materials response. In this analysis we have chosen to focus on an energetic approach in which we compute the accumulated kinetic energy transmitted to the blade leading edge instead of using approaches based on the waterhammer equation that seek to compute the impact pressure and material response to the resulting Rayleigh, shear and compression waves (that are assumed to act independently from each individual impact) (Slot et al., 2015; Dashtkar et al., 2019). It is important to reiterate that the approach adopted here i.e. to compute the maximum total kinetic energy transferred to the blade, which is used here as a proxy

15  for the erosion potential, represents the upper bound on actual kinetic energy transfer since it employs a closing velocity characteristic for the tip of WT rotors, assumes all falling hydrometeors impact the blade, and neglects energy loss during the transfer, 'splash' and bounce of hydrometeors. There are more complex frameworks that can be applied to simulate the pressure and transient stresses on the blade coatings (Mishnaevsky Jr, 2019) and impingement erosion (Amirzadeh et al., 2017a, b). A model of the blade response to precipitation impacts could be incorporated within the analysis framework to examine the

20  probability and time to exceed the (Cumulative) Failure Threshold Energy (Fiore et al., 2015).

This work suggests the dominance of hail as a damage vector for WT blades at all of the sites studied here. This is consistent with indications that deep convection and hail are particularly common in the central US (Cintineo et al., 2012) and of large geographic variability in hail frequency (Ni et al., 2017). This finding emphasizes the key importance of efforts to build and enhance hail climatologies (Allen et al., 2015; Gagne et al., 2019) with applications in a wide range of industries (from insurance

25  to renewable energy). The dominance of hail as a damage vector and the importance of a relatively small number of 5-minute periods to total annual kinetic energy transfer from rain adds credence to the proposal that blade LEE could be greatly reduced by operating erosion-safe turbine control (Bech et al., 2018) wherein the WT are curtailed during periods with extreme precipitation (very heavy rain or the occurrence of hail) without substantial loss of income.

**5**    **Acknowledgments**

30    This research was funded by the US Department of Energy (DE-SC0016438) and Cornell University's Atkinson Center for a Sustainable Future (ACSF-sp2279-2018). It was enabled by access to computational resources supported via the NSF Extreme Science and Engineering Discovery Environment (XSEDE) (award TG-ATM170024) and ACI-1541215. The authors gratefully acknowledge the scientists and technicians of the National Weather Service for their work in realizing the dual polarization RADAR network and making the data publicly available. We appreciate the contributions of our two peer reviewers in making

[revised manuscript text omitted]

10  Gaudern, N.: A practical study of the aerodynamic impact of wind turbine blade leading edge erosion, Journal of Physics: Conference Series, 2014, 012031,

Giguère, P., and Selig, M. S.: Aerodynamic effects of leading-edge tape on aerofoils at low Reynolds numbers, Wind Energy, 2, 125-136, 1999.

Herring, R., Dyer, K., Martin, F., and Ward, C.: The increasing importance of leading edge erosion and a review of existing
15  protection solutions, Renewable and Sustainable Energy Reviews, 115, 109382, 2019.

Heymsfield, A. J., Giammanco, I. M., and Wright, R.: Terminal velocities and kinetic energies of natural hailstones, Geophysical Research Letters, 41, 8666-8672, 2014.

[revised manuscript text omitted]